# Conserved function of the HAUS6 calponin homology domain in anchoring augmin for microtubule branching

**Martin Würtz** [1,2,9] ✉, **Giulia Tonon** [1,7,9], **Bram J. A. Vermeulen** [1], **Maja Zezlina**[1], **Qi Gao**[1], **Annett Neuner**[1], **Angelika Seidl**[3], **Melanie König** [4], **Maximilian Harkenthal** [1], **Sebastian Eustermann** [2], **Sylvia Erhardt** [3,5], **Fabio Lolicato** [6,8], **Elmar Schiebel** [1] & **Stefan Pfeffer** [1] ✉

Branching microtubule nucleation is a key mechanism for mitotic and meiotic spindle assembly and requires the hetero-octameric augmin complex. Augmin recruits the major microtubule nucleator, the γ-tubulin ring complex, to pre-existing microtubules to direct the formation of new microtubules in a defined orientation. Although recent structural work has provided key insights into the structural organization of augmin, molecular details of its interaction with microtubules remain elusive. Here, we identify the minimal conserved microtubule-binding unit of augmin across species and demonstrate that stable microtubule anchoring is predominantly mediated via the calponin homology (CH) domain in Dgt6/HAUS6. Comparative sequence and functional analyses in vitro and in vivo reveal a highly conserved functional role of the HAUS6 CH domain in microtubule binding. Using cryo-electron microscopy and molecular dynamics simulations in combination with AlphaFold structure predictions, we show that the *D. melanogaster* Dgt6/HAUS6 CH domain binds microtubules at the inter-protofilament groove between two adjacent β-tubulin subunits and thereby orients augmin on microtubules. Altogether, our findings reveal how augmin binds microtubules to pre-determine the branching angle during microtubule nucleation and facilitate the rapid assembly of complex microtubule networks.

Microtubules (MTs) are fundamental components of the eukaryotic cytoskeleton, playing crucial roles in diverse cellular processes such as compartmentalization, chromosome segregation, cell motility, and intracellular transport. At the onset of division, cells assemble a dense and polarized spindle composed of interconnected MT networks, including pole-to-pole, pole-to-kinetochore, and branched MTs, to ensure the precise segregation of the genetic material[1–4]. The MT branching mechanism is governed by the coordinated interplay of many different components, most notably the γ-tubulin ring complex (γ-TuRC), the primary MT nucleator[5–9], and the augmin complex, which recruits the γ-TuRC to pre-existing MTs[1]. Through this bridging function, the augmin complex determines the orientation and polarity of

[1]Zentrum für Molekulare Biologie der Universität Heidelberg (ZMBH), Heidelberg, Germany. [2]European Molecular Biology Laboratory (EMBL), Heidelberg, Germany. [3]Zoological Institute, Karlsruhe Institute of Technology (KIT), Karlsruhe, Germany. [4]Biochemie-Zentrum der Universität Heidelberg (BZH), Heidelberg, Germany. [5]Institute of Biological and Chemical Systems - Functional Molecular Systems (IBACS-FMS), Karlsruhe Institute of Technology (KIT), Karlsruhe, Germany. [6]Department of Physics, University of Helsinki, Helsinki, Finland. [7]Present address: Department of Molecular Sociology, Max Planck Institute of Biophysics, Frankfurt, Germany. [8]Present address: Biochemie-Zentrum der Universität Heidelberg (BZH), Heidelberg, Germany. [9]These authors contributed equally: Martin Würtz, Giulia Tonon. ✉e-mail: m.wuertz@zmbh.uni-heidelberg.de; s.pfeffer@zmbh.uni-heidelberg.de

newly formed MTs[10–12] and therefore is of utmost importance for the rapid amplification of spindle MTs, as well as for establishing connections with kinetochore fibers[13,14]. Consistent with these central functions, loss of augmin results in mitotic and meiotic spindle assembly defects, chromosome missegregation, and ultimately cell lethality[15–20]. Beyond its spindle-associated roles, augmin is essential for non-centrosomal MT networks in neurons, where it supports axonal MT organization, influences MT polarity, and plays an important role in central nervous system development[21–27].

Recent structural studies shed light on the molecular architecture of the augmin complex[28–31] revealing that its previously reported h-shaped structure[32,33] is formed by two tetrameric subcomplexes (Fig. 1a): the rod-like TIII subcomplex (composed of HAUS1, HAUS3, HAUS4, and HAUS5) and the clamp-like TII subcomplex (composed of HAUS2, HAUS6, HAUS7, and HAUS8). These two subcomplexes serve complementary functions, providing the binding sites for the γ-TuRC (TIII) and the pre-existing MT (TII), respectively[33–35] (Fig. 1a). While the interaction between TIII and the γ-TuRC remains elusive, the N-terminal half of TII, termed N-clamp, was proposed to function as the MT binding site[28,29,32,33]. In particular, the unstructured positively charged N-terminal extension of the TII subunit HAUS8 (HAUS8-NT) was suggested to bind the negatively charged MT surface, thereby serving as an essential MT-binding element of human augmin[32,36]. Yet, interaction data suggest that additional parts of TII are required to establish a stable and effective interaction[32], and the defined angle at which MT branching occurs is challenging to reconcile with a MT binding site formed purely by the disordered HAUS8-NT.

Recent structural analysis of vertebrate augmin revealed that the HAUS8-NT extends from a hammerhead-like structure comprising two additional potential MT-binding elements: the globular calponin homology (CH) domains of HAUS6 and HAUS7[28–31], hereafter referred to as CH6 and CH7, respectively (Fig. 1b). CH domains adopt a compact, globular fold consisting of four main alpha-helices and form the MT-binding motifs in numerous MT-associated proteins, like the MT plus-end binding EB proteins[37,38], the kinetochore proteins Ndc80 and Nuf2[39,40] and the inner centriole protein CEP44[41]. In these complexes, the CH domains are often accompanied by unstructured extensions, which typically play an important role in the ability of the protein to bind MTs[40,42,43]. Per similarity, the composition of the N-clamp thus suggests that augmin binds MTs through a composite interface involving the unstructured HAUS8-NT and one or two of the CH domains[28–31].

Here, we elucidate the molecular basis for the augmin-MT interaction by reconstituting the augmin N-clamp as a minimal MT-binding unit across various model organisms. We identify the low-complexity MT-binding unit from *D. melanogaster* as an optimal model for dissecting the central function of CH6 in MT binding. By integrating cryo-EM analysis, in silico modeling and in vitro assays with in vivo data, we characterize how CH6 anchors augmin to the inter-protofilament groove between adjacent β-tubulin subunits. Our study reveals a conserved MT-binding mechanism utilized by augmin to achieve initiation of branched MT formation and proper spindle assembly.

## Results

### Augmin N-clamp is a structurally conserved MT-binding unit
To identify the minimal structurally conserved MT-binding unit of augmin, we analyzed augmin TII structures predicted by AlphaFold2 (AF2)[44,45] across a range of model organisms. Our comparative analysis, including vertebrates (*X. laevis* and *H. sapiens*), invertebrates (*D. melanogaster*) and plants (*A. thaliana*), revealed that the general architecture of augmin TII as predicted by AF2 is remarkably conserved (Fig. 1c, Supplementary Fig. 1a–c). The predicted TII architecture is characterized by extensive coiled-coil bundles, bisected by a hinge region that provides conformational flexibility between the N- and C-clamp, consistent with previous studies[28–31]. Notably, the augmin

N-clamp in *D. melanogaster* included only a single CH domain formed by the HAUS6 ortholog Dgt6 (hereafter named Dgt6$_{HAUS6}$), and lacked the second CH domain mapping at the N-terminus of Msd5$_{HAUS7}$ in other organisms[28,30]. Additionally, we observed substantial length and charge variability in the unstructured N-termini of HAUS8 orthologs (Fig. 1c, Supplementary Fig. 2), especially for the *D. melanogaster* Dgt4$_{HAUS8}$-NT, which is the shortest amongst those considered and contains no positively charged side chains.

To experimentally test and compare the MT binding function of augmin N-clamp from these key model organisms and validate the AF2 structure predictions, we established structure-guided recombinant reconstitution of the minimal augmin MT-binding units from *H. sapiens*, *X. laevis*, *A. thaliana*, and *D. melanogaster* in *E. coli*.

We co-expressed the augmin N-clamp components based on standardized dual expression constructs (*HAUS2*, *HAUS6*, *HAUS7-EGFP-HIS$_8$*, and *HAUS8-HIS$_8$*) (Supplementary Fig. 3a). Using His-affinity purification combined with size exclusion chromatography (SEC) and anion exchange chromatography (AEC), we successfully purified augmin N-clamp from all indicated organisms (Supplementary Fig. 3b), as also confirmed by mass spectrometry (MS) analysis (Supplementary Tables 1–4).

Negative stain electron microscopy (EM) analysis and 2D class averaging verified complex formation and validated general features of the AF2 structure predictions (Fig. 1c, Supplementary Fig. 3c). The 2D class averages recapitulated all ordered structural elements of the augmin N-clamp complexes, including two globular densities representing the CH domains of HAUS6 and HAUS7 orthologs, along with an elongated density segment corresponding to the extended coiled-coil region formed by HAUS2, 6, 7 and 8. As predicted, only a single CH domain was resolved in the *D. melanogaster* augmin N-clamp. This was further reflected in the reduced molecular weight of the *D. melanogaster* augmin N-clamp compared to human augmin N-clamp in mass photometry experiments (Supplementary Fig. 3d). Due to its flexibility, the N-terminal extension of HAUS8 orthologs was not resolved by negative stain EM analysis.

To assess the function of the purified augmin N-clamp complexes in MT binding, we conducted tubulin co-sedimentation assays, an established method for characterizing the MT binding properties of proteins. All recombinant augmin N-clamp complexes demonstrated MT binding activity in vitro, co-pelleting with polymerized α/β-tubulin in a concentration-dependent manner (Fig. 1d, e). Importantly, human augmin N-clamp bound to MTs with an affinity comparable to that reported for the full TII subcomplex[32] (Table 1, Fig. 1d, e), indicating that the augmin N-clamp is a fully functional minimal MT-binding unit.

Notably, the bacterial expression system does not replicate the extensive post-translational modifications that augmin subunits undergo during mitosis. As a result, the properties of the augmin N-clamp observed in biochemical assays with recombinant augmin N-clamp may represent only a subset of its properties in vivo. However, our results demonstrate that the augmin N-clamp is a highly suitable, low-complexity model for dissecting the interaction between augmin and MTs in vitro from both mechanistic and structural perspectives.

### The role of HAUS8-NT for MT binding varies between species
Next, we used our recombinant in vitro system (Supplementary Fig. 4a) combined with tubulin co-sedimentation assays to systematically dissect the contribution of different augmin N-clamp components to MT binding. The unstructured HAUS8-NT of human augmin was reported to be essential for the augmin-MT interaction[32,36]. Consistently, deletion of the HAUS8-NT in the human augmin N-clamp completely abolished MT binding (Fig. 2a, Supplementary Fig. 4b, Table 1). While MT binding was also impaired in both *X. laevis* and *A. thaliana* augmin N-clamp constructs lacking the HAUS8-NT (Fig. 2b, c, Supplementary Fig. 4b, Table 1), they still exhibited concentration-dependent MT-binding activity, albeit with reduced apparent binding affinities and

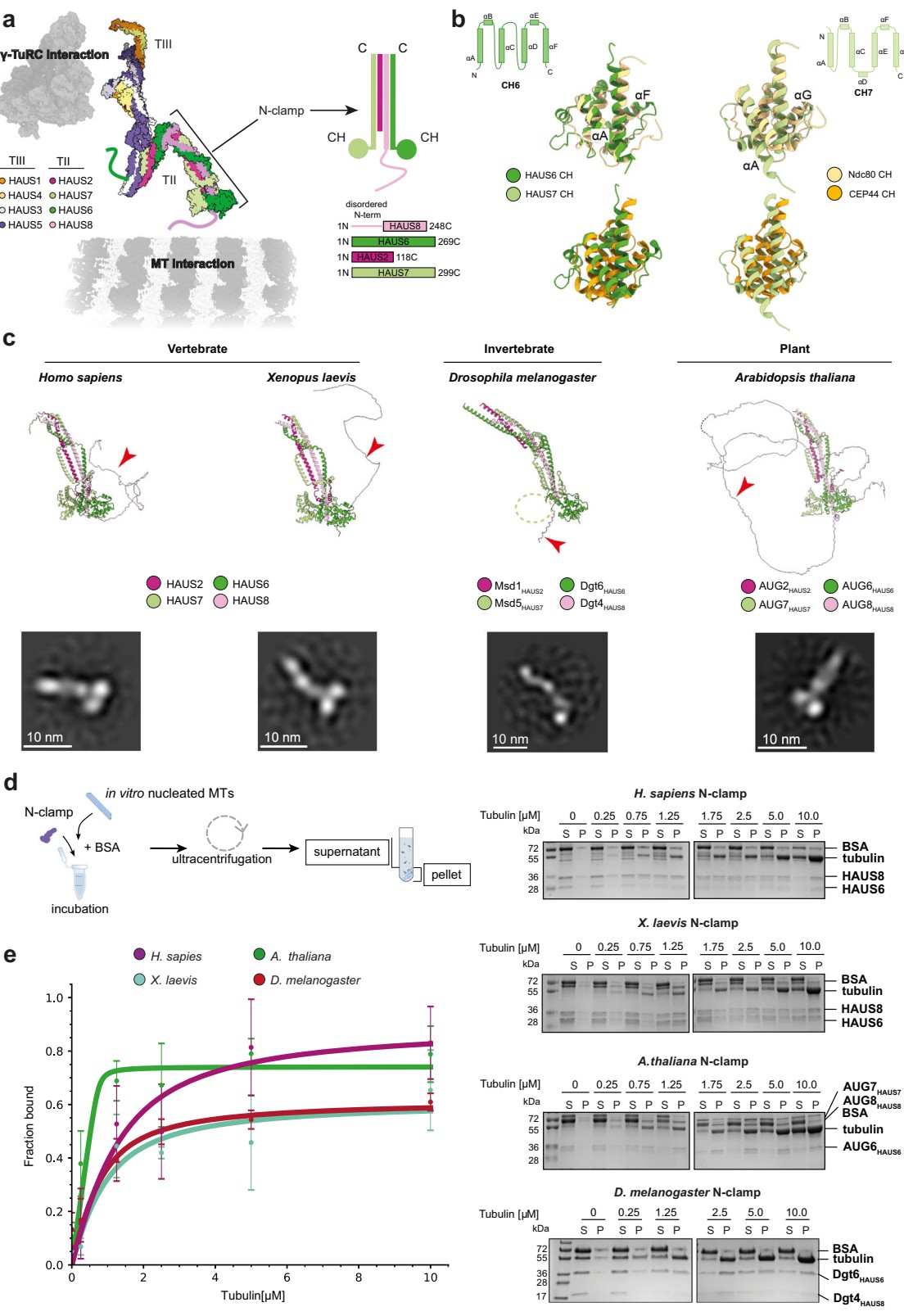

maximum binding capacities. In contrast, a much milder effect was observed upon HAUS8-NT deletion in *D. melanogaster* (Fig. 2d, Supplementary Fig. 4b, Table 1), where the Dgt4$_{HAUS8}$-NT is substantially shorter than in other species (Supplementary Fig. 2) and completely devoid of the positively charged residues that have been postulated to drive MT binding in its vertebrate counterparts[32,36]. Overall, these data reveal a differential effect of the HAUS8-NT on MT binding in different species, likely reflecting evolutionary adaptations of the MT branching machinery.

## The CH domain of HAUS6 is a conserved MT-binding element

Having tested the role of HAUS8-NT across species, we next turned our attention to the CH6 and CH7 domains, first focusing on structural conservation and electrostatic charge distribution. Generally, surface

**Fig. 1 | Augmin N-clamp is a structurally conserved minimal MT-binding unit.** **a** Schematic representation of the augmin holocomplex (left)[29] and a cartoon illustrating the augmin N-clamp architecture (right, *H. sapiens*). Coloring as indicated. **b** Organization of the CH domains in HAUS6 and HAUS7. Middle: Overlay of augmin's CH domains with CH domains from other MT-binding proteins. Shown are human proteins: HAUS6, HAUS7 (PDB-7SQK), Ndc80 (PDB-3IZ0) and CEP44 (PDB-7PT5). **c** Top: Top-ranked AF2 models of the augmin N-clamp from different species, with coloring as indicated (see Supplementary Fig. 1). The unstructured region of the HAUS8-NT is indicated with a red arrowhead. The missing second CH domain (CH7) domain for *D. melanogaster* is indicated with a circle. Bottom**:** Representative negative stain EM 2D classes of purified augmin N-clamp from different species. **d** Left: Schematic representation of tubulin co-sedimentation assay. Right: Representative SDS-PAGE analyses of tubulin co-sedimentation assays for wild-type augmin N-clamp from different species. Tubulin concentration is indicated on the top. Supernatant (S) and Pellet (P) fractions are indicated (see Supplementary Fig. 3). **e** Pelleted fraction from tubulin co-sedimentation assays for different species plotted against the tubulin concentration. Shown as mean ± SD (see Methods and Table 1) with fitted binding curves, coloring as indicated. Experiments were performed in at least *n* = 3 repetitions: *H. sapiens n* = 5; *X. laevis n* = 3; *D. melanogaster n* = 4; *A. thaliana n* = 3. Source data are provided as a Source Data file.

## Table 1 | Quantification of MT binding affinity for wild-type and mutated augmin N-clamp complexes from different species as determined using tubulin co-sedimentation assays

| Sample | $K_d$ [µM] | SD of $K_d$ | $R^2$ | Number of experiments |
|---|---|---|---|---|
| H. sapiens[wild-type] | 0.8 | 0.4 | 0.94 | 5 |
| X. laevis[wild-type] | 0.6 | 0.4 | 0.91 | 3 |
| A. thaliana[wild-type] | 0.01 | 0.09 | 0.89 | 3 |
| D. melanogaster[wild-type] | 0.4 | 0.3 | 0.90 | 4 |
| D. melanogaster[Msd5-GST] | 0.04 | 0.01 | 0.90 | 3 |
| H. sapiens[HAUS8-ΔNT] | – | – | – | 3 |
| X. laevis[HAUS8-ΔNT] | 2.2 | 2.1 | 0.69 | 3 |
| A. thaliana[AUG8-ΔNT] | 2.1 | 1.5 | 0.89 | 3 |
| D. melanogaster[Dgt4- ΔNT] | 2.9 | 0.5 | 0.99 | 3 |
| H. sapiens[HAUS6-4A-mut] | 5.6 | 1.8 | 0.98 | 3 |
| D. melanogaster[Dgt6-3A-mut] | 25.5 | 31.9 | 0.94 | 3 |
| D. melanogaster[Msd5- Δ81–93] | 1.0 | 1.0 | 0.61 | 3 |

Apparent $K_d$ values, standard deviation (SD) and $R^2$ of the fitted curve (see Methods, Figs. 1–3) are shown. Dgt4/AUG8 and Dgt6 are the *D. melanogaster/A. thaliana* orthologs of HAUS8 and HAUS6, respectively. Number of experiments is given. No binding was observed for *H. sapiens*[HAUS8-ΔNT](-).

residues on CH6 and CH7 were more conserved on the surface distal than proximal to the coiled-coil region (Fig. 3a), suggesting that the distal surface is more likely to participate in a conserved function like MT binding. Interestingly, while the CH7 surface is predominantly negatively charged, the CH6 surface is largely basic (Fig. 3b). This indicates a higher potential for CH6 to mediate interaction with the overall negatively charged surface of MTs, which is also consistent with the complete absence of CH7 in *D. melanogaster*. Consistently, sequence alignment of HAUS6 orthologs across eukaryotes (Supplementary Fig. 5a), from plants to human, identified several conserved basic residues on CH6, specifically K84, R90, K91, K98, and K123 (human residue numbers), forming a solvent-exposed basic patch (Fig. 3b) that may underlie specific CH6-MT interactions to orient augmin on MTs.

To test this prediction, we introduced point mutations in human CH6 (K84A, R90A, K91A, K98A, referred to as HAUS6[4A-mut], Supplementary Fig. 5b) and performed tubulin co-sedimentation assays with wild-type and mutant human augmin N-clamp. We observed a sevenfold reduction in MT binding affinity for HAUS6[4A-mut] compared to human wild-type HAUS6 (Fig. 3c, Supplementary Fig. 5c, Table 1). The residual MT binding ability likely originates from the HAUS8-NT. These data suggest that while the HAUS8-NT of human augmin is essential for initial binding, CH6 likely is required for stably anchoring augmin to MTs.

To test the function of the conserved CH6 residues in a low-complexity system without interference from other MT-binding N-clamp elements, we analyzed *D. melanogaster* augmin N-clamp, where the CH7 domain is absent and the Dgt4[HAUS8]-NT deletion had little impact on MT binding (Fig. 2d). Strikingly, mutating three of the conserved residues in *D. melanogaster* Dgt6[HAUS6] (K82A, R88A, and K96A; referred to as Dgt6[HAUS6][3A-mut]) resulted in a much more severe impairment in augmin N-clamp-MT binding (Fig. 3d, Supplementary Fig. 5b, c, Table 1) compared to the human HAUS6[4A-mut] N-clamp variant. This indicates that the conserved basic patch in the CH6 domain plays an important role in the augmin-MT interaction, which is more critical for establishing high-affinity MT binding in *D. melanogaster* than in human augmin.

Finally, to confirm the role of the CH6 domain in MT binding in *D. melanogaster* in vivo, we generated stable S2 cell lines expressing either *Dgt6[HAUS6][WT]-GFP* or *Dgt6[HAUS6][3A-mut]-GFP* under the control of the metallothionein promoter. Endogenous Dgt6[HAUS6] was depleted using Dgt6[HAUS6] dsRNA (3 days or 4 days), and metaphase cells expressing either wild-type or mutant *Dgt6[HAUS6]* were analyzed by immunofluorescence staining (Fig. 3e, Supplementary Fig. 6a–d). Dgt6[HAUS6][3A-mut]-GFP exhibited clearly reduced colocalization with metaphase spindles compared to Dgt6[HAUS6][WT]-GFP (Fig. 3e, f, Supplementary Fig. 6d, e), which localized to spindles as expected for the functional augmin complex[18].

These results establish the CH6 domain as a conserved MT-binding element across species, linking in silico analysis with both in vitro and in vivo functional data.

### CH6 directly binds to MTs to orient augmin on the MT lattice

Next, we aimed at elucidating the structural basis for how CH6 anchors and orients augmin on MTs using cryo-EM. To this end, we used the *D. melanogaster* augmin N-clamp as a low-complexity model system, in which CH6 plays a dominant role in the augmin-MT interaction, while the CH7 domain is absent and the Dgt4[HAUS8]-NT deletion had little impact on MT binding. Pre-incubation of taxol-stabilized MTs (see Methods) with *D. melanogaster* augmin N-clamp resulted in pronounced MT bundling complicating cryo-EM analysis (Supplementary Fig. 7a–c). However, sequential application of MTs and augmin N-clamp onto cryo-EM grids yielded individual MTs decorated with *D. melanogaster* augmin N-clamp, suitable for cryo-EM analysis (Supplementary Fig. 7d, e). Following established cryo-EM image processing workflows (see Methods)[46], we reconstructed the 14-protofilament MT to ~4 Å resolution at the tubulin level (Fig. 4a, Supplementary Fig. 8a–e), enabling us to unambiguously distinguish α- and β-tubulin (Fig. 4b). Inspection of the cryo-EM reconstruction at low density threshold level identified the augmin N-clamp binding site in the groove between β-tubulin subunit pairs on adjacent protofilaments (Fig. 4a). Subsequent symmetry expansion and 3D classification focused on the asymmetric unit provided a cryo-EM reconstruction of the augmin N-clamp bound to the MT (Supplementary Fig. 8f, g). However, the limited local resolution for the decorating density (Supplementary Fig. 8f) and the absence of density attributable to more distal regions of the augmin N-clamp prevented confident docking of the N-clamp model predicted by AF2.

To improve the cryo-EM sample quality, we next aimed at boosting the overall N-clamp decoration of MTs by enhancing the

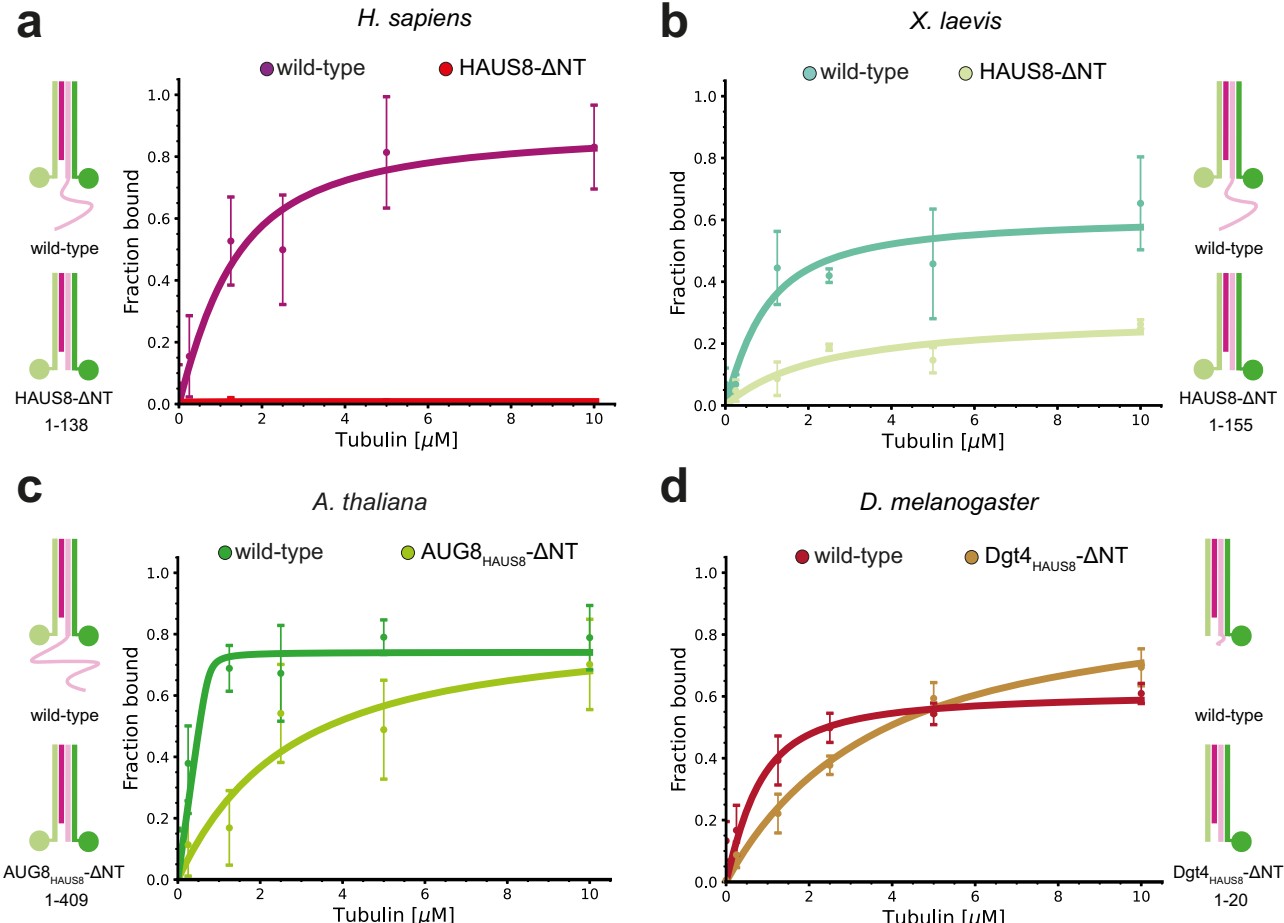

**Fig. 2 | The HAUS8-NT differentially contributes to MT binding across species.**
**a-d** Pelleted fraction from tubulin co-sedimentation assays for wild-type and
HAUS8-ΔNT augmin N-clamp variants from different species plotted against
tubulin concentration. Schematic icons indicate the organization of the corresponding augmin N-clamp and the deleted amino acids of HAUS8-NT. Shown as
mean ± SD (see Methods and Table 1) with fitted binding curves, coloring as indicated. Curves of the wild-type complexes from Fig. 1. Experiments with HAUS8-ΔNT
constructs were performed in $n = 3$ repetitions. Source data are provided as a
Source Data file.

N-clamp binding affinity, which is relatively low (~0.4 μM) compared to
similar MT-binding complexes successfully analyzed by cryo-EM, such
as Ndc80-related constructs[40]. To preserve the native binding mode,
we aimed at increasing binding affinity by GST-mediated dimerization
of the augmin N-clamp. We placed a GST tag after the C-terminal EGFP
tag on Msd5HAUS7, separated by a flexible linker (Supplementary
Fig. 9a), generating a GST-tagged *D. melanogaster* augmin N-clamp
version (GST-N-clamp). Complex formation and GST-induced dimerization were confirmed by mass photometry (Supplementary Fig. 9b,
c). Tubulin co-sedimentation assays showed that GST-mediated
dimerization decreased the apparent $K_d$ approximately tenfold to
~40 nM (Supplementary Fig. 9d, e). We repeated cryo-EM experiments
with the GST-N-clamp (see Methods) analogously to wild-type N-clamp
(Supplementary Fig. 10a–d). After assigning the α/β-tubulin register
(Fig. 4c), we observed decorating density at the same position as for
wild-type N-clamp, confirming the MT binding site of augmin. Compared to our cryo-EM reconstruction of the wild-type augmin N-clamp,
the local estimated resolution for the GST-N-clamp was substantially
improved (Supplementary Fig. 10b) and the more distal coiled-coil
regions of the GST-N-clamp were better visible at low threshold level
(Supplementary Fig. 10c). Subsequent recentering of particles on the
asymmetric unit, combined with additional rounds of focused 3D
classification and 3D refinement, further improved the cryo-EM
reconstruction to secondary structure level for the entire CH6
domain and MT-proximal segments of Msd5HAUS7 and Dgt4HAUS8

(Fig. 4d, Supplementary Fig. 10d, e). This enabled highly confident
rigid body docking of the augmin N-clamp model predicted by AF2 and
revealed the binding site and orientation of the augmin N-clamp on
MTs (Fig. 4d, e). In this configuration, the conserved basic patch in the
augmin CH6 domain is positioned to directly bind to the MT surface,
indicating that it centrally contributes to orienting augmin on the MT
lattice. Notably, MT-proximal α-helices of Msd5HAUS7 and Dgt4HAUS8
were slightly differently oriented between the cryo-EM reconstruction
and the AF2 prediction, which may reflect local inaccuracies of the
model predicted by AF2 or suggest small conformational rearrangements of the augmin N-clamp upon MT binding.

**MD simulations identify central interacting residues**
To assess the stability of the interface and gain residue-level insight
into the interactions involved, we conducted atomistic Molecular
Dynamics (MD) simulations of the composite *D. melanogaster* augmin N-clamp–MT model (see Methods). We simulated three independent replicas for 1 μs each and analyzed the final 200 ns and
500 ns of each simulation. Throughout all simulations, augmin
N-clamp remained stably bound at the inter-protofilament groove
between two adjacent β-tubulin subunits (see Supplementary
Movie 1, Supplementary Fig. 11a). Residue-level contacts between the
augmin N-clamp and surrounding tubulin subunits were analyzed
within a 0.6 nm cutoff and averaged across the three replicas. The
average contact frequencies of augmin N-clamp with β-tubulin

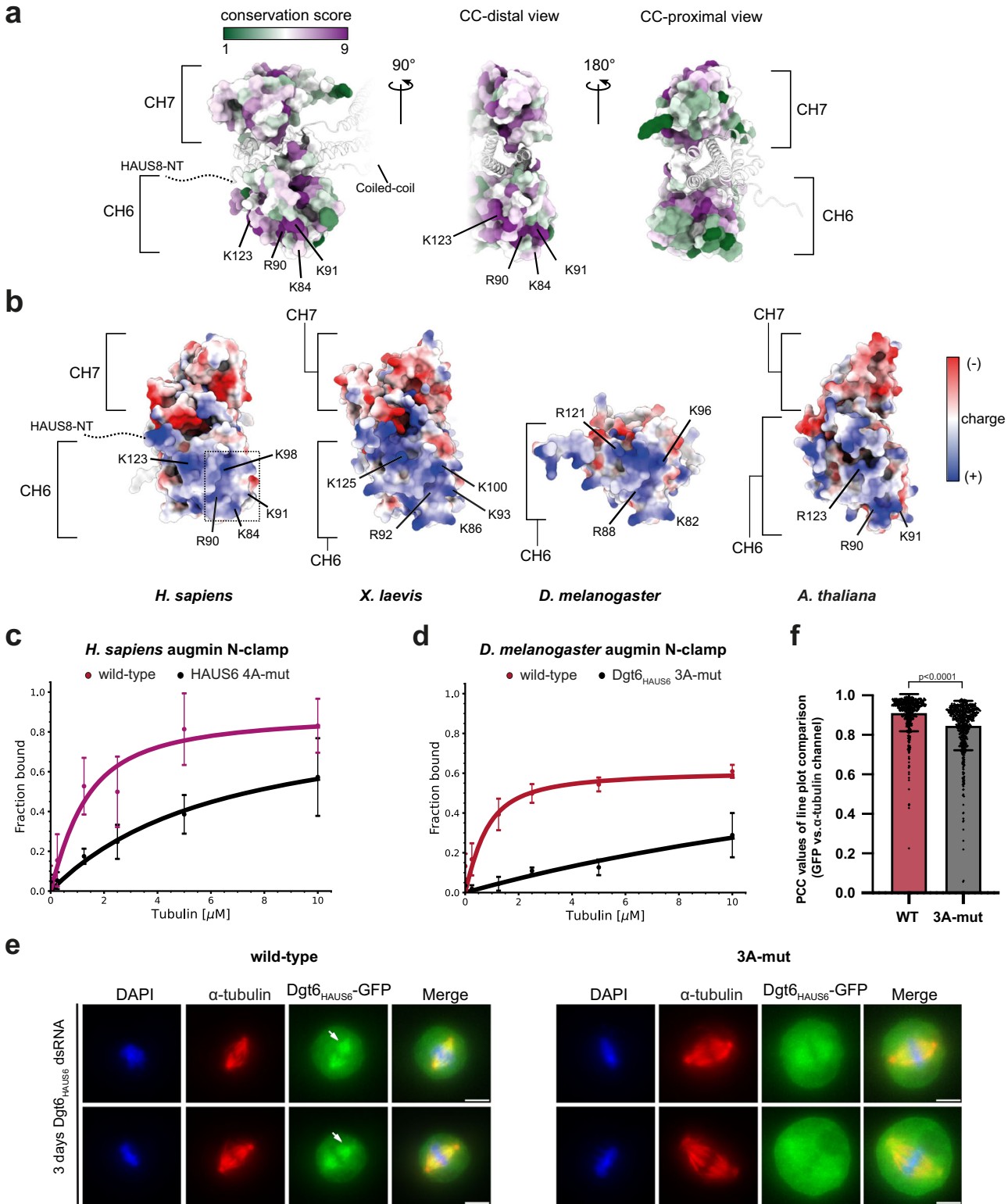

indicated that the majority of interactions were driven by the CH6 domain and are located in the regions of residues 5-13, 71-88 and 110-121 (Fig. 5a, b, Supplementary Fig. 11b, c; Supplementary Fig. 12a, b). Additionally, we identified a flexible loop region in Msd5$_{HAUS7}$ (residues 85–92) that strongly interacted with the disordered C-termini of β-tubulin rich in acidic residues, known as the E-hook (Fig. 5a, b, Supplementary Fig. 11b, c; Supplementary Fig. 12a, b; Supplementary Fig. 13a, b)[47]. This loop in Msd5$_{HAUS7}$ appears to be fly-specific and correlates with the absence of the CH7 domain and reduced length

and charge in the Dgt4$_{HAUS8}$-NT in these species (Supplementary Fig. 14a). Notably, our cryo-EM density shows unassigned density segments extending from the tubulin E-hook region adjacent to the N-clamp binding site (Supplementary Fig. 10f), aligning with MD-simulation data that suggest E-hook contacts contribute to augmin N-clamp binding in *D. melanogaster*.

To refine the simulation-based contact map, we calculated the interaction energy between residues, highlighting the contributions of both short- and long-range interactions (Fig. 5a, Supplementary

**Fig. 3 | CH6 is a conserved MT-binding element. a** Model of the human augmin N-clamp hammerhead region (PDB-7SQK) colored according to conservation score, shown from viewing angles distal and proximal to the coiled-coil (CC) region. Coiled-coil region and HAUS8-NT are indicated. Coloring scheme and conserved residues are indicated. **b** Augmin N-clamp hammerhead region from human (PDB-7SQK), *X. laevis* (PDB-8AT3), and AF2 predictions for *D. melanogaster* and *A. thaliana* were colored according to charge distribution. View distal to the coiled-coil region (see Supplementary Fig. 5a). **c** Pelleted fraction from tubulin co-sedimentation assays for wild-type and HAUS6[4A-mut] (K84A, R90A, K91A, K98A) human augmin N-clamp variants plotted against tubulin concentration. Shown as mean ± SD with fitted binding curves (see Methods and Table 1), coloring as indicated. Curve of wild-type complex from Fig. 1. Experiments were performed in n = 3 repetitions. **d** Pelleted fraction from tubulin co-sedimentation assays for wild-type and Dgt6[HAUS6][3A-mut] (K82A, R88A, K96A) N-clamp variants from *D. melanogaster* plotted against tubulin concentration. Shown as mean ± SD with fitted binding

curves (see Methods and Table 1), coloring as indicated. Curve of wild-type complex from Fig. 1. Experiments were performed in n = 3 repetitions. **e** Immuno-fluorescence images following dsRNA-mediated Dgt6[HAUS6] knockdown in *D. melanogaster* S2 cells. Cells expressing either Dgt6[HAUS6][wild-type]-GFP or Dgt6[HAUS6][3A-mut]-GFP constructs were stained for α-tubulin, DAPI, and GFP (Dgt6[HAUS6]). Representative sections of images showing metaphase cells from wild-type *(left)* and 3A-mut *(right)*. Scale bars: 5 μm. Clear localization to spindle MTs is indicated with white arrowheads in case of wild-type (see Supplementary Fig. 6). **f** Quantification of Dgt6[HAUS6] spindle localization after Dgt6[HAUS6] dsRNA treatment (3 or 4 days, see Supplementary Fig. 6). The intensity values of eight line plots per spindle (GFP vs α-tubulin) were compared using Pearson correlation coefficients (PCC) and PCC values were plotted for WT = 68 cells (purple) and 3A-mut = 75 cells (gray). n = 4 experiments (see Supplementary Fig. 6e). The bars represent mean ± SD, statistical significance is indicated by p-values determined using two-sided Mann–Whitney *U* test. Source data are provided as a Source Data file.

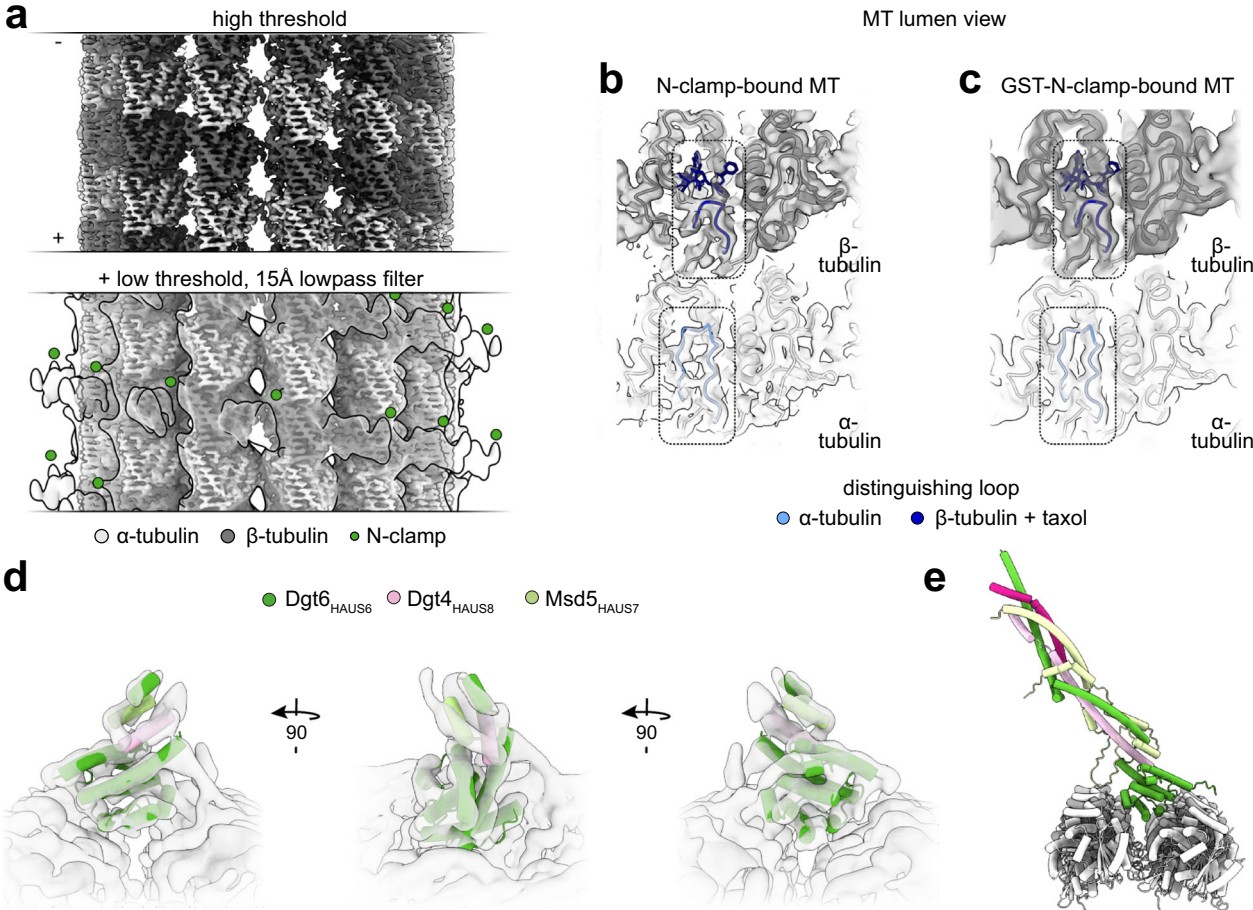

**Fig. 4 | CH6 orients *D. melanogaster* augmin N-clamp on MTs. a** Helical reconstruction of a 14-PF MT decorated with the *D. melanogaster* augmin N-clamp shown at high confidence threshold (top) and overlaid with the same density at lower confidence threshold (transparent) after low-pass filtering to 15 Å (bottom). Green circles indicate additional density in the groove between neighboring β-tubulin subunits. Coloring as indicated. **b** MT-lumenal view of one α/β-tubulin dimer in the cryo-EM density from (a) superposed with the atomic model of an α/β-tubulin dimer (PDB-6WVR)[97]. Loop S9-S10 region, distinguishing α- and β-tubulin, is outlined and shown in blue along with taxol (stick representation); otherwise, coloring

as in panel (**a**). **c** Same as (**b**), but showing the reconstruction of the 13 PF MT decorated with GST-N-clamp shown in Supplementary Fig. 10c. **d** 3D reconstruction of 13PF MT-bound augmin GST-N-clamp after focused 3D classification of symmetry-expanded particles, filtered to local resolution. The AF2 prediction of *D. melanogaster* augmin N-clamp was rigid-body docked. Distal segments of the AF2 prediction are not shown. Coloring as indicated. **e** AF2 prediction of the full *D. melanogaster* augmin N-clamp placed on the MT lattice based on the N-clamp fit shown in (**d**). Coloring as in (**d**).

Fig. 13a). This analysis indicated that the interaction network between augmin N-clamp and β-tubulin is primarily driven by electrostatic interactions and ion pairs (Fig. 5a, b, Supplementary Fig. 13a–c). Key residues, making frequent interactions throughout both time ranges include R3, K10, K48, K81, K82, R88 and R121 of Dgt6[HAUS6], as well as

K86, K87 and R88 in Msd5[HAUS7]. In contrast to β-tubulin, α-tubulin plays a minor role in interacting with the augmin N-clamp. Only α-tubulin E-hook residues D431 and E434 of α-tubulin form strong interactions with R3 and the N-terminus of Dgt6[HAUS6] (Supplementary Fig. 15a–f).

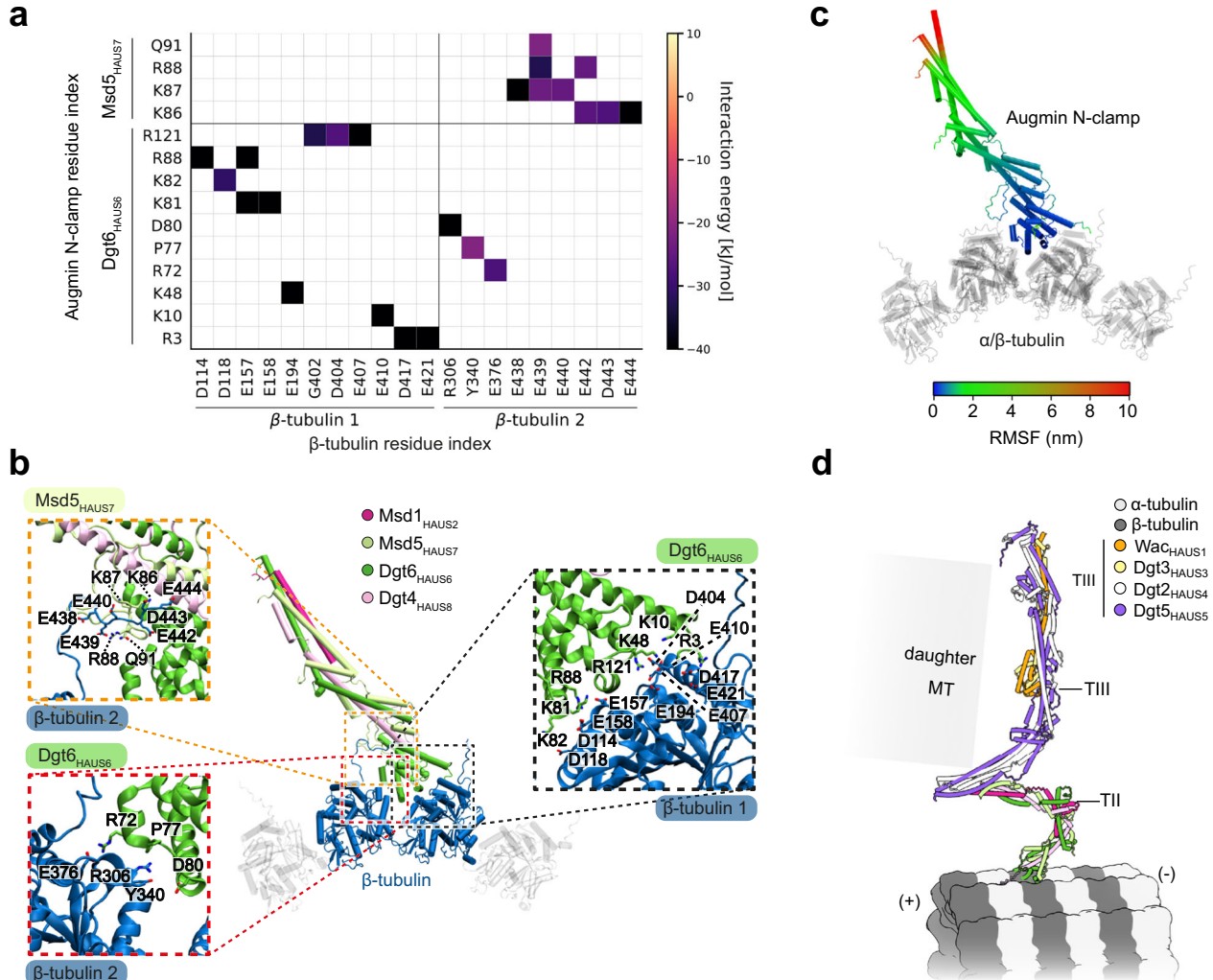

**Fig. 5 | Molecular characterization of the augmin N-clamp-MT interface via atomistic molecular dynamics simulations. a** Pairwise interaction energy map highlighting augmin N-clamp residues interacting with the two β-tubulin monomers, displaying the average interaction energy as the sum of electrostatic and van der Waals contributions, based on the contacts identified in Supplementary Fig. 12. Key interacting residues with an interaction energy of less than −20 kJ/mol are shown. **b** Representative structure of the MT-bound augmin N-clamp, with close-up views (red, black, orange) highlighting all residues identified as involved in contacts. Key interacting residues with an interaction energy of less than -20 kJ/mol are labeled, see Supplementary Fig. 12b. **c** Root-Mean-Square-Fluctuation (RMSF) of the MT-bound augmin N-clamp, with the color gradient representing the degree of fluctuations (in nm) along the structure. Blue regions indicate minimal fluctuation, while red regions emphasize higher flexibility. The RMSF is calculated for the entire trajectory of each run and subsequently averaged across all runs, coloring is indicated. **d** Orientation of the full *D. melanogaster* augmin complex placed on the MT, based on a composite AF2 prediction of the augmin octamer (see Supplementary Fig. 16). TII and TIII tetramers, MT polarity and possible daughter MT direction are indicated. Coloring of TII as in (**b**), TIII as indicated.

To further investigate the contribution of the Msd5$_{HAUS7}$ loop region in MT binding, we deleted the Msd5$_{HAUS7}$ loop residues forming strong interactions in the MD simulations (K86, K87, and R88; Msd5$_{HAUS7}^{Δ81−93}$) from the *D. melanogaster* N-clamp. Following purification, complex integrity of the mutant N-clamp was confirmed by mass photometry (Supplementary Fig. 14b–d). MT co-sedimentation assays demonstrated that mutant Msd5$_{HAUS7}^{Δ81−93}$ N-clamp was strongly compromised in MT binding (Supplementary Fig. 14e, f, Table 1), validating an important role of the Msd5$_{HAUS7}$ loop region in MT binding and thereby supporting a model in which this loop may substitute for the HAUS8-NT conserved in other eukaryotes.

Next, we calculated the root-mean-square-fluctuation (RMSF) to assess the dynamics of the MT-bound augmin N-clamp. While CH6 segments in contact with the MT are highly rigid, consistent with the strong network of electrostatic interactions, fluctuations increase significantly as they propagate away from the MT (Fig. 5c, Supplementary Fig. 13c), which may explain why only the MT-bound region of

the N-clamp could be resolved using cryo-EM at high density confidence threshold. Overall, these data are fully consistent with our structural, functional and evolutionary analysis (Fig. 3, Fig. 4), further underlining the critical role of residues in the CH6 basic patch for MT binding.

Finally, we investigated how the orientation of the *D. melanogaster* augmin N-clamp consistently observed in MD simulations and cryo-EM structural analysis positions the full augmin complex on MTs. Using a "divide-and-conquer" approach adapted from a previously published protocol[29], we modeled the complete *D. melanogaster* augmin complex based on AF2 predicted models (Supplementary Fig. 16a–d). Combining these predictions with the observed MT binding mode of the *D. melanogaster* N-clamp, we generated a composite model of augmin bound to MTs (Fig. 5d). Anchoring via CH6 aligns the γ-TuRC-binding augmin TIII subcomplex along the MT axis, oriented toward the MT plus end. In this configuration, augmin TIII is sterically highly accessible to interact with γ-TuRCs for nucleation of daughter MTs.

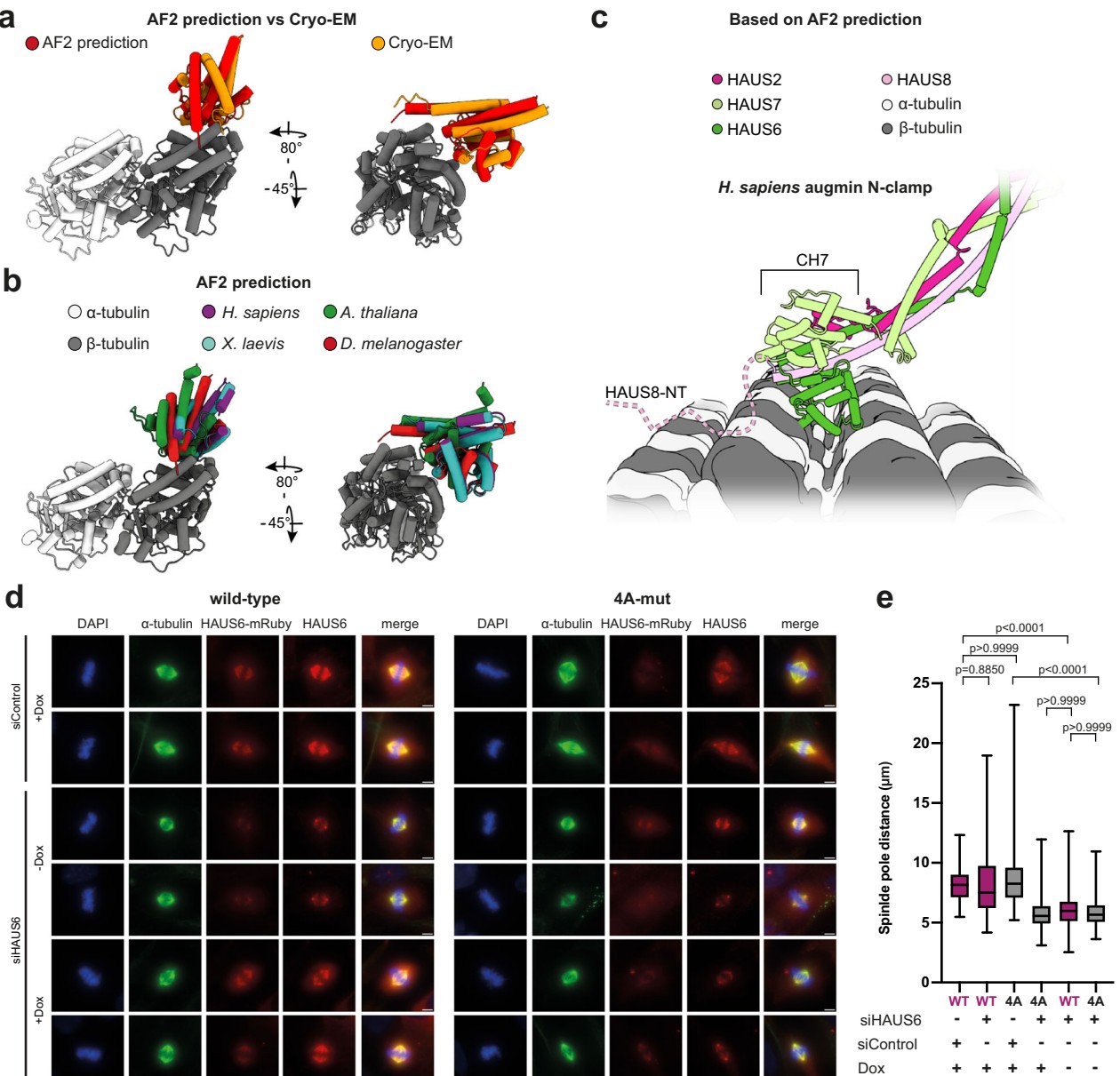

**Fig. 6 | CH6 binding is conserved across species and compatible with complex augmin N-clamp architecture. a** Top-ranked AF2 predictions of α/β-tubulin (porcine) in complex with the *D. melanogaster* CH6 domain compared to the cryo-EM model of GST-N-clamp bound to MT (Fig. 4e), presented in two views. Coloring as indicated. **b** Top-ranked AF2 predictions of α/β-tubulin (porcine) in complex with the CH6 domains from four different species, presented in two views. For prediction metrics and sequences used for the predictions see Supplementary Fig. 19 and Supplementary Table 7, respectively. **c** Human augmin N-clamp (PDB-7SQK) placed on the MT based on the AF2 predicted position of CH6 shown in (b). Potential direction of HAUS8-NT binding is indicated. **d** Representative immuno-fluorescence images of pRetroX-Tet-On RPE1 cells expressing siRNA-resistant *HAUS6^wild-type^*-mRuby (WT) and *HAUS6^4A-mut^*-mRuby (4A-mut) in metaphase. Cells were treated with either siRNA against HAUS6 (siHAUS6) or a non-specific control siRNA (siControl). Expression of *HAUS6*-mRuby constructs was induced by 20 ng/ml doxycycline (+Dox) or left uninduced (-Dox). Cells were stained for α-tubulin, DAPI, and HAUS6, and imaged as well for HAUS6-mRuby channel. Scale bars: 5 μm. **e** Spindle-pole-to-spindle-pole distance was measured for all conditions in wild-type (WT, purple) and 4A-mut (4A, gray) cells. The data are presented as boxplots with median as central line and 25th and 75th percentile as borders of the box with minimum and maximum values as whiskers. Sample sizes for each condition: WT siControl, *n* = 102 cells; WT siHAUS6, *n* = 136 cells; 4A siControl, *n* = 158 cells; 4 A siHAUS6 *n* = 187 cells; WT siHAUS6 (−Dox), *n* = 166 cells; 4 A siHAUS6 (−Dox), *n* = 130 cells. Data were pooled from three independent experiments. Statistical significance is indicated by *p*-values determined using two-sided Kruskal–Wallis test with Dunn's multiple comparison test. Source data are provided as a Source Data file.

## CH6-MT interactions are likely conserved

Next, we aimed to understand whether the CH6 binding mode experimentally observed for *D. melanogaster* augmin N-clamp was evolutionarily conserved and compatible with the more complex augmin N-clamp architecture in other organisms. We utilized AF2 to predict an α/β-tubulin dimer in complex with the *D. melanogaster* CH6 domain. In 15 out of 25 models, the position of the *D. melanogaster* CH6 domain matched our cryo-EM reconstruction (Fig. 6a). The small deviations between the predicted and experimental position of CH6 most likely arise from the absence of the second β-tubulin subunit that the CH6 domain binds to in the cryo-EM reconstruction. This high similarity between the predicted and experimental models of the CH6-MT complex in *D. melanogaster* indicated that AF2 is generally capable of identifying correct MT-CH6 interactions. Therefore, we extended

our AF2 analysis of CH6-MT interactions to *H. sapiens*, *X. laevis*, and *A. thaliana* CH6 domains (Fig. 6b, Supplementary Fig. 17a, b). In all top-ranked models, CH6 was predicted to bind β-tubulin in an orientation similar to that observed in *D. melanogaster*, suggesting a conserved binding mode for CH6 domains in all inspected organisms (Fig. 6b).

Superposing the full human augmin N-clamp[30] onto the human CH6 domain as positioned on the MT surface by AF2 (Fig. 6c) revealed that the more complex N-clamp architecture in vertebrates and plants, including a longer HAUS8-NT and the additional CH7 domain, is fully compatible with the CH6 orientation predicted by AF2 and experimentally determined by cryo-EM in case of *D. melanogaster*. This further supports a conserved binding mode for CH6 across species.

To test whether the conserved and predicted CH6-MT interface is functionally relevant in human augmin, we generated pRetroX-Tet-On RPE1 cell lines expressing either siRNA-resistant wild-type or 4A-mutant *HAUS6-mRuby* (Supplementary Fig. 18a, b) and tested the impact of the CH6-4A mutations under siRNA-mediated knockdown of endogenous *HAUS6* (Fig. 6d, Supplementary Fig. 18c). Expression of *HAUS6-mRuby* constructs was confirmed via immunoblotting (Supplementary Fig. 18b) and immunofluorescence analysis demonstrated robust centrosome localization during interphase and MT localization during cell division, consistent with previous findings[17,18,48] (Supplementary Fig. 18d, e). By immunoblotting, we verified knockdown of endogenous *HAUS6*, but not siRNA-resistant *HAUS6-mRuby*, upon siRNA treatment and doxycycline-induced expression (Supplementary Fig. 18f–h). Previous studies have established that knockdown of augmin components results in severe mitotic spindle defects, most prominently manifested in shortened spindles[13,14]. Consistently, siRNA-mediated knockdown of *HAUS6* (siHAUS6) resulted in visibly impaired spindles (Fig. 6d), characterized by a significant reduction in spindle-pole-to-spindle-pole distance (Fig. 6e). Wild-type *HAUS6-mRuby* expression rescued the spindle length defects, restoring spindle dimensions to near-wild-type levels. In contrast, expression of the 4A-mutant phenocopied *HAUS6* knockdown and resulted in significantly shorter mitotic spindles (Fig. 6e), demonstrating that the conserved residues in CH6 are critical for the function of human augmin. Collectively, these results suggest that the MT-CH6 interactions characterized here represent a conserved mechanism essential for correct binding of augmin to spindle MTs and the proper assembly of the mitotic spindle.

## Discussion

In this study, we identify how augmin stably binds to MTs via the N-clamp, a structurally conserved composite MT-binding unit shared across plants, invertebrates and vertebrates. Our data establish that the conserved CH6 domain plays a critical role in MT binding throughout all species studied. We focused our study on *D. melanogaster*, where CH6 is central for the augmin-MT interaction by binding to the inter-protofilament groove between two β-tubulins, as demonstrated by a combination of cryo-EM structure determination, AF2 predictions and MD simulations, in conjunction with in vitro tubulin co-sedimentation and in vivo spindle co-localization assays. These findings considerably advance our mechanistic understanding of augmin's composite MT-binding interface, which was hypothesized to involve a second critical contact site in addition to the HAUS8-NT[32,36] over a decade ago, which we now identify as the CH6 domain.

Interestingly, we found that the requirement for HAUS8-NT in MT binding varies between species. While its deletion in *D. melanogaster* has the mildest effect, the much more extended HAUS8-NT in humans is essential for MT binding, likely due to its positively charged residues that interact with the negatively charged MTs through ionic interactions[32,36,49,50]. Surprisingly, in *X. laevis* and *A. thaliana*, HAUS8-NT deletion compromises but not completely abolish MT binding, underscoring the important role of HAUS6 in augmin´s MT binding affinity and function. This aligns with experimental data

demonstrating that the human HAUS8-NT alone binds MTs with substantially lower efficiency than the full augmin complex or the augmin TII tetramer[32,49]. In light of our results, this indicates that while HAUS6/Dgt6 is universally important for MT binding, the necessity of HAUS8-NT in recruiting and stabilizing the CH6-MT interaction is species-dependent. In case of *D. melanogaster*, our MD simulations and tubulin co-sedimentation assays for instance indicate that the MT binding function of the very short Dgt4$_{HAUS8}$-NT could be partially replaced by the disordered loop of Msd5$_{HAUS7}$, which engaged tubulin E-hook residues through a patch of positively charged residues (K86, K87, R88, R92) in the simulations − reminiscent of the K-loop of KIF1A[47,51]. Overall, this suggests that stable anchoring of CH6 may generally require additional electrostatic interactions, provided for example by the positively charged human HAUS8-NT or the *D. melanogaster* Msd5$_{HAUS7}$ loop. In some instances, posttranslational modifications such as phosphorylation may fine-tune those electrostatic interactions, as reported for vertebrate and plant HAUS8-NT[52–54].

Although we provide insights into the functional interplay between CH6 and HAUS8-NT, the role of CH7 remains unclear. The appearance of multiple CH domains in a protein complex is not uncommon, especially in actin-binding proteins, where one CH domain often assumes a stabilizing or regulatory function, while the other mediates binding[55,56]. The conserved negative charge distribution in CH7 domains suggests a conserved mechanism of action, perhaps in recruiting interaction partners or balancing augmin's electrostatics to enable augmin clustering at MT branching sites (see below)[12].

MT branching angles have been analyzed across species, including *H. sapiens*, *X. laevis*, *D. melanogaster*, and plants, revealing context-dependent patterns. In human in vitro systems and *Xenopus* egg extracts, branching nucleation mediated by γ-TuRC and augmin predominantly occurs at shallow angles below 30°, reflecting the dense MT packing necessary for spindle assembly and function[2,10,12]. In plants, branching angles have been characterized within cortical arrays and phragmoplasts, typically ranging from 20° to 60°[57]. In *D. melanogaster*, both in vitro and in vivo analyses reveal branching angles typically ranging from 30° to 60°[11,58]. While the unstructured HAUS8-NT is unlikely to provide a uniform and stable platform for orienting the γ-TuRC on pre-existing MTs, the conserved CH6 domain and its defined binding site on the MT surface are a prime candidate to fulfill this function. Consistently, while the human augmin N-clamp with CH6-4A mutations still bound to MTs in pelleting assays due to the contribution of HAUS8-NT, functional analyses showed severely disrupted spindle function and assembly in cells expressing *HAUS6$^{4A-mut}$*, as witnessed by reduced pole-to-pole distance, a phenotype that is typical for loss of augmin function[13,14]. How regulatory and augmin-cooperating factors may affect the branching angle by modulating the augmin-MT interaction remains to be investigated in molecular detail. For example, the plant protein MACET4 has been shown to affect MT branching angles and MT organization by modulating binding of the augmin TII subunit AUG7$_{HAUS7}$ to MTs, highlighting the potential for additional regulatory layers to fine-tune augmin anchoring to MTs[59].

Similar to the cooperativity observed in Ndc80 complexes and other multivalent assemblies, where clustering amplifies both binding affinity and functional specificity[60,61], a recent study suggested a potential clustering mechanism of augmin at MT branching sites[12]. In light of our structural data, clustering of multiple augmin complexes per branch site could involve binding of several copies of augmin to β-tubulin subunit pairs in close spatial proximity. Augmin's conformational flexibility[29], in particular in the hinge region separating the augmin TII N- and C-clamp, may play a central role in enabling γ-TuRC interactions from MT-binding sites with a slight offset. In vertebrates, augmin clusters may also involve multiple copies of augmin, which are not stably docked to the MT lattice, but only flexibly tethered by the

HAUS8-NT. Key posttranslational modifications and augmin-interacting branching factors like γ-TuRC, HURP, NEDD1 or TPX2[12,62,63] may facilitate or further assist augmin clustering.

Notably, we observed pronounced MT bundling activity of our N-clamp construct in vitro (Supplementary Fig. 7), consistent with previous reports for HAUS8 (formerly HICE1)[36]. Although MT bundle formation has been reported in vivo in the context of kinetochore fibers and dense spindle arrays[13], whether augmin directly contributes to this organization through bundling activity is unknown. In fact, MT bundling is a common feature of multivalent MT-binding proteins in vitro[64,65] but may arise from high local protein concentrations or other non-physiological experimental conditions. Addressing whether the MT bundling effect of the augmin N-clamp reflects a specific biological function will require further investigations.

Apart from augmin, the MT-binding protein TPX2 has also been described as a pivotal component of MT branching in *X. laevis* and is able to recruit augmin and γ-TuRC to MTs[10,62]. Cryo-EM analysis has revealed the TPX2 binding site in the inter-protofilament groove between two adjacent β-tubulin units close to the CH6 binding site[66], suggesting that direct N-clamp-TPX2 interactions may underlie the augmin-TPX2 cooperativity during MT branching[9,52].

In sum, by characterizing the N-clamp as a minimal MT-binding unit of augmin and demonstrating the conserved role of CH6 in orienting augmin on MTs, we provide a basis for exploring species-specific regulatory mechanisms, additional MT binding partners, and evolutionary adaptation of augmin. In particular, investigating augmin's interplay with the γ-TuRC, TPX2, or HURP[62,63] will be essential for uncovering the complexity of its regulation and its indispensable role in cell division.

## Methods

### Molecular cloning
For each cloning step, the vectors were linearized and amplified by PCR (Q5, NEB). All PCR-amplified fragments were assembled using the NEBuilder Assembly Kit following the manufacturer's instructions (NEB). DNA fragments of *HAUS2*, *HAUS6*, *HAUS7*, and *HAUS8* from different species (Supplementary Table 5) were cloned into two separate *E. coli* expression plasmids based on the pETDuet-1 and pET-26b expression vectors using primers listed in Supplementary Table 6.

For *H. sapiens* and *X. laevis*, in two subsequent cloning steps, *HAUS8* and *HAUS6* encoding sequences were inserted into the two pETDuet-1 cloning sites, adding a C-terminal 8xHis tag to the *HAUS8* sequence. *HAUS2* and *HAUS7* were inserted into another pETDuet-1 vector. Afterwards, the pET-26b expression cassette was replaced with the pETDuet-1 expression cassette including the gene fragments of *HAUS2* and *HAUS7* using primers listed in Supplementary Table 6. During this step, the sequence of *TEV-EGFP-3C-8xHis* tag was added to the C-terminus on the *HAUS7* fragment using a pET26b backbone as described[29]. Gene fragments from *A. thaliana* and *D. melanogaster* were codon-optimized for *E. coli* expression (IDT, USA) and ordered as gene fragments (TWIST Bioscience) for direct insertion into the pET-Duet-1/pET-26b vector scheme replacing the corresponding gene fragments. For the generation of EGFP-GST tagged Msd5$_{HAUS7}$ the GST sequence was amplified from a pGEX plasmid and pET-26b Msd1$_{HAUS2}$/Msd5$_{HAUS7}$ plasmid was amplified with primers listed in Supplementary Table 6.

For protein expression, these constructs were used for co-transformation. Alternatively, to enhance transformation and expression efficiency, the two expression cassettes of pETDuet-1 and pET-26b were combined into the pETDuet-1 vector. To achieve this, the pET-Duet1 backbone containing the *HAUS6* and *HAUS8* gene fragments, and the gene cassette from the modified pET-26b vectors were amplified separately. In this step, the 8xHis tag following *HAUS7-TEV-EGFP* was removed by replacing it with a stop codon. These

components were then combined, resulting in a pETDuet-1 vector containing all four gene cassettes (Supplementary Fig. 3a).

Mutations and deletions in the constructs were introduced by PCR amplification followed by DpnI (NEB) digestion and transformation into DH5α *E. coli*, as for the other constructs.

Full-length pMT *Dgt6$_{HAUS6}$$^{WT}$-GFP* and *Dgt6$_{HAUS6}$$^{3A-mut}$-GFP* constructs for expression in *D. melanogaster* were cloned into the pMT-GFP expression vector. For this purpose, the remaining C-terminal segment of *Dgt6$_{HAUS6}$* was ordered as a gene fragment (TWIST Bioscience), including a C-terminal TEV-ALFA tag, and then cloned together with the N-terminal segment of *Dgt6$_{HAUS6}$* into the GFP-V5 region of the pMT plasmid (see Supplementary Fig. 6a).

pRetroX-Tet3G and pVSVG (Retro-X Tet-On 3G Inducible Expression System, Clontech) were used to generate the RPE1 cells with Tet-On 3 G System. The pRetroX-Tet3G with *HAUS6* WT or 4A-mut constructs were cloned sequentially. First, non-coding base-exchange mutations for siRNA resistance were introduced into the pETDuet-1 *HAUS6* N-clamp constructs. Next, point mutation PCR was performed to incorporate the 4A mutations using primers listed in Supplementary Table 6. Then, the siRNA-resistant wild-type or mutated N-terminal part of *HAUS6* was cloned into a full-length *HAUS6* construct (pACEBac1 HAUS6-2xFLAG). Finally, the full-length *HAUS6* from the pRetroX-Tet3G vector and the mRuby sequence flanked by ALFA-tag were amplified and combined via a NEBuilder reaction.

### Protein expression
For the expression of augmin N-clamp constructs, 100-150 ng of DNA constructs were transformed into 100 µl of competent *E. coli* BL21 CodonPlus-RIL (Stratagene). Transformed cells were cultured in 2xYT medium (Roth) at 37 °C in volumes up to 500 ml until an optical density (OD) of maximal 0.8 was reached. The expression cultures were then shifted to 18 °C and diluted with an equal volume of cold 2xYT medium. Protein expression was induced at an OD of 0.6-0.8 by adding 0.5 mM IPTG, and the cultures were incubated at 18 °C for 20-21 h with shaking. Cells were harvested by centrifugation (4000*g*, 4 °C for 20 min), washed with PBS (Gibco), and either used immediately for protein purification or flash-frozen in liquid nitrogen and stored at −80 °C until further use.

### Protein purification
For the purification of recombinant augmin N-clamp complexes, expression pellets from 1 liter culture were resuspended in 30–35 ml cold lysis buffer (20 mM HEPES, pH 7.4, 300 mM KCl, 1 mM MgCl$_2$, 1 mM EGTA, 2 mM DTT, 0.1% Tween-20), supplemented with one EDTA-free protease inhibitor tablet (Roche) and 10 µl Benzonase (Sigma Aldrich). Resuspended cells were kept on ice and sonicated (6 × 1 min at 100% amplitude; Hielscher UP50H). The lysate was clarified by centrifugation (20,000*g* for 30 min at 4 °C). The total soluble fraction of the lysate was incubated with Ni-NTA beads (Qiagen) pre-equilibrated with lysis buffer for 90 min at 4 °C with gentle rotation. Typically, 400 µl of Ni-NTA beads were used per 1 l of culture. After incubation, beads were separated from the flow-through by centrifugation (800*g*, 3 min at 4 °C) and washed once with lysis buffer, followed by two washes with wash buffer (20 mM HEPES, pH 7.4, 200 mM KCl, 15 mM imidazole, 1 mM MgCl$_2$, 1 mM EGTA, 1 mM DTT). After each washing step, the beads were sedimented by centrifugation (800*g*, 3 min at 4 °C). Recombinant proteins were eluted three times from the Ni-NTA beads by incubating with 1–1.25 bead volumes of elution buffer (20 mM HEPES, pH 7.4, 200 mM KCl, 400 mM imidazole, 1 mM MgCl$_2$, 1 mM EGTA, 1 mM DTT) for 10 min at 4 °C. Elution fractions were collected by centrifugation (800*g*, 3 min at 4 °C) and subjected to SEC using a Superose 6 Increase (10/300) GL column equilibrated in SEC buffer (20 mM HEPES, pH 7.4, 150 mM KCl, 1 mM MgCl$_2$, 1 mM EGTA, 1 mM DTT, 2.5% glycerol) at 0.5 ml/min. Peak

fractions were verified by Coomassie-stained SDS-PAGE analysis (4-20% gradient SDS Precast-gel; BIO-RAD), pooled, and subjected to AEC. GST-N-clamp was used for cryo-EM experiments and mass photometry after SEC. For AEC, peak fractions from SEC were loaded onto a Capto HiRes™ Q 5/50 GL column (Cytiva) equilibrated with low salt buffer (20 mM HEPES, pH 7.4, 75 mM KCl, 1 mM $MgCl_2$, 1 mM EGTA, 1 mM DTT, 2.5% glycerol). Recombinant N-clamp complexes were eluted using a gradient from 75 mM to 1 M KCl at a flow rate of 0.5 ml/min over 20 column volumes. All chromatography runs were performed on an ÄKTA Pure system or ÄKTA go system (Cytiva) controlled by Unicorn software (version 7.9 or 7.5). Peak fractions were verified via Coomassie-stained SDS-PAGE (4–20% gradient SDS Precast-gel; BIO-RAD), combined, and concentrated using Amicon concentrators (30 kDa MWCO). The purified proteins were used directly or flash-frozen in liquid nitrogen and stored at −80 °C until further use. Porcine tubulin was purified according standard protocols as previously described[67].

## Stable RPE1 cell lines
Human telomerase-immortalized RPE1 cells (hTERT-RPE1 TP53−/−) were cultured in Dulbecco's Modified Eagle's Medium (DMEM)/F-12 (Gibco) supplemented with 10% FBS, 1% penicillin–streptomycin, and 1% L-glutamine. Cells were maintained at 37 °C in a 5% $CO_2$-humidified incubator. Stable cell lines were firstly generated by introducing the Retro-X™ Tet-On® 3 G Inducible Expression System (Clontech) into the RPE1 $TP53^{-/-}$ cells. Different HAUS6 siRNA-resistant constructs were then integrated under the TRE3G promotor via Retrovirus infection (Clontech).

## RNA interference and rescue in RPE1 cells
Endogenous HAUS6 depletion was achieved by transfecting cells with siRNA oligonucleotides targeting HAUS6 (5′-CAGUUAAGCAGGUAC-GAA-3′) (Dharmacon™)[48] using Lipofectamine RNAiMAX (Invitrogen). Transfection reactions were prepared in Opti-MEM™ medium according to the manufacturer's protocol. RPE1 cells were seeded to 60–80% confluency in 6-well plates. siRNA oligonucleotides (50 nM) were incubated with RNAiMAX in serum-free Opti-MEM medium (Gibco) for 20 min, then added to the cells. After 24 h, cells were reseeded into 24-well plates containing 12 mm coverslips for another 48 h. To rescue the depletion and to analyze the function, the HAUS6 wild-type or 4A-mut construct expression were induced by adding 20 ng/ml of doxycycline to the cell medium 48 h before the end of the siRNA depletion experiment.

## D. melanogaster S2 cell transfection
Sterile culture conditions were maintained for D. melanogaster S2 cells at 25 °C. The culture medium consisted of Schneider's medium supplemented with 10% heat-inactivated fetal bovine serum (FBS) and 100 μg/ml Penicillin/Streptomycin. The expression of the Dgt6$_{HAUS6}$-GFP wild-type and 3A-mut gene, driven by the metallothionein promoter (pMT vector), was induced using $CuSO_4$ (500 μM for 16 h). D. melanogaster S2 cells stably expressing pMT constructs were generated via Cellfectin II mediated transfection (Life Technologies™), with hygromycin used as the selection marker. Two days post-transfection, the medium was supplemented with 200 μg/ml hygromycin B. The medium was refreshed weekly. After 4 weeks, once the cells resumed exponential growth, the hygromycin concentration was reduced to 100 μg/ml.

## dsRNA-mediated knockdown
Double-stranded RNA (dsRNA) was synthesized using the MEGAscript T7 kit (Ambion) following the manufacturer's protocol. Templates were generated by PCR from cDNA using specific primers for Dgt6$_{HAUS6}$ and brown (control). The sequences of primers and dsRNA are listed in Supplementary Table 6. Knockdown efficiency was assessed via qPCR.

A total of $6 \times 10^5$ logarithmically growing S2 cells were seeded in 12-well plates, with 2 ml of medium per well. On the following day, cells were incubated for 1 h with 8 μg of dsRNA in 0.5 ml of serum-free medium. Next, 1 ml of medium containing 20% serum was added, and cells were incubated for 3 or 4 days. 16 h before harvesting, the Dgt6$_{HAUS6}$-GFP construct was induced with 500 μM $CuSO_4$. Cells were used for immunofluorescence (IF) or RNA extraction.

## RNA qPCR
RNA isolation was performed using Trizol® reagent according to the manufacturer's protocol. Any remaining genomic DNA was digested using TURBO™ DNase (Ambion by Life Technologies). cDNA synthesis was carried out using the Quantitect Reverse Transcription Kit (Qiagen). A control reaction without reverse transcriptase was included to detect potential gDNA contamination. qPCR was performed in triplicates on a LightCycler 480 multiwell plate, using LightCycler 480 SYBR Green I Master (Roche) in combination with the LightCycler 480 instrument (Roche). The primer pair PP14341 for quantifying Dgt6$_{HAUS6}$ was obtained from FlyPrimerBank. Its efficiency was tested and measured at 97%. dGapdH served as the reference gene for normalization. The qPCR data were analyzed using the $2^{-\Delta\Delta CT}$ method to determine relative gene expression levels[68]. Wild-type S2 cells were treated with 8 μg of Dgt6$_{HAUS6}$ dsRNA, while the control group received brown dsRNA. Harvesting was performed at 3- and 6-day post-treatment.

## Immunofluorescence
For D. melanogaster S2 cells, a 100 μl cell suspension was placed on a Poly-L-lysine-treated glass coverslip. Cells were allowed to settle for 10 min before fixation with 4% paraformaldehyde (PFA) for an additional 10 min. After three washes with PBS, cells were permeabilized for 7 min using PBS containing 0.3% Triton X-100. Nonspecific binding was blocked by incubating cells for 1 h with 4% BSA in PBS. All antibodies were diluted in blocking solution and incubated for 1 h at room temperature. GFP was stained using Thermo Fisher A10262 antibody (1:200, Thermo Fisher), and α-tubulin was stained with Sigma T9026 antibody (1:700). After three washes with PBS, cells were incubated with the corresponding Alexa Fluor secondary antibodies (1:500), protected from light. After another three washes with PBS, cells were mounted in ProLong™ Diamond Antifade Mountant with DAPI (Invitrogen™) on a glass slide.

For experiments in RPE1 cells, HAUS6 wild-type or 4A-mut cells were grown on 12 mm coverslips and fixed with cold methanol at −20 °C for 5 min. They were then blocked with PBS supplemented with 10% FBS and 0.1% Triton X-100 for 1 h at room temperature (RT). Afterwards, cells were incubated with primary antibody, diluted in 3% BSA solution, for 1 h. Following primary antibody incubation, cells were stained with secondary antibody and DAPI in 3% BSA solution for 45 min, then mounted with Moviol. Image acquisition was performed using a DeltaVision RT system (Applied Precision) with an Olympus IX71 microscope, equipped with 60×/1.42 and 100×/1.40 oil objective lenses.

Metaphase cells from the Drosophila S2 cell experiment were analyzed to compare Dgt6$_{HAUS6}$$^{wild-type}$-GFP and Dgt6$_{HAUS6}$$^{3A-mut}$-GFP cells following Dgt6$_{HAUS6}$ dsRNA treatment. For each metaphase spindle, four lines were drawn through each spindle pole side, resulting in eight lines in total (Supplementary Fig. 6). Signal intensity along these lines in all channels was measured using the line plot tool in Fiji software[69]. The intensity values between the GFP and tubulin channels for each line were compared by calculating the Pearson correlation coefficient (PCC) between the two channels. The data of 3 or 4 days Dgt6$_{HAUS6}$ dsRNA treatment were compared and plotted individually (Supplementary Fig. 6, as well as pooled in Fig. 3f). For the RPE1 experiments, spindle pole distances were measured using the Fiji software[69].

## Antibodies

Primary antibodies used in this study for immunofluorescence (IF) and immunoblot were: GFP was stained using A10262 chicken antibody (1:200, Thermo Fisher). In *D. melanogaster* S2 cells α-tubulin was stained with Sigma T9026 mouse antibody (1:700), whereas for RPE1 cells mouse α-tubulin (monoclonal, clone 1E4C11, Proteintech, IF 1:500, REF 66031-1IG, LOT 10028345) was used. Alpaca HRP-conjugated sdAb anti-ALFA (monoclonal, clone 1G5, NanoTag Biotechnologies GmbH, WB 1:1000, REF N1505-HRP, LOT 032303B), rabbit GAPDH (monoclonal, clone 14C10F5G9, Cell Signaling Technology, WB 1:1000, REF #2118, LOT 16), rabbit polyclonal antibodies against HAUS6 (polyclonal, AA 448–955, WB 1:2000, IF 1:1000)[70].

Secondary HRP-conjugated antibody used in immunoblot was HRP-conjugated anti-rabbit (polyclonal, Jackson ImmunoResearch, WB 1:5000, REF 711-035-152, 172401). Secondary antibodies used for immunofluorescence were: Alexa Fluor 488 goat anti-chicken (Thermo Fisher Scientific, A-11039, 1:500), Alexa Fluor 647 goat anti-mouse (Thermo Fisher Scientific, A-21235, 1:500), Alexa Fluor Plus 488 donkey anti-mouse IgG (polyclonal, Thermo Fisher Scientific, IF 1:500, REF A-11001, LOT 2816171), Alexa Fluor Plus 647 donkey anti-rabbit IgG (polyclonal, Thermo Fisher Scientific, IF 1:500, REF A32733, LOT 2577247).

## LC-MS/MS analysis

Each aliquot of human, *X. laevis*, *A. thaliana*, and *D. melanogaster* augmin N-clamp was thawed and clarified by centrifugation at 20,000*g* for 5 min at 4 °C. The clarified samples were then mixed with an equal volume of 2x Lämmli buffer and boiled at 95 °C for 3 min. Protein separation was carried out by SDS-PAGE (minigel, 10%; Invitrogen). Coomassie-stained bands were excised with a scalpel and processed as previously described[71,72]. Briefly, the samples were reduced, alkylated, and digested with trypsin. Peptides were extracted from the gel pieces, concentrated using a vacuum centrifuge, and dissolved in 15 µl of 0.1% TFA.

Nanoflow LC-MS/MS analysis of the four distinct augmin N-clamp samples was conducted utilizing two distinct LC-MS platforms: an Ultimate 3000 liquid chromatography (LC) system coupled to an Orbitrap Q Exactive HF (QE HF; Thermo Fisher Scientific), and a Vanquish Neo coupled to an Orbitrap Tribrid Eclipse mass spectrometer (Thermo Fisher Scientific). In both configurations, peptide separation was performed on an in-house packed analytical column (75 µm × 200 mm) using Reprosil-Pur AQ 120 C18 stationary phase material with 1.9 µm particle size (Dr. Maisch, Germany). Each sample analysis was performed with a single biological replicate ($n = 1$).

For analyses performed using the Ultimate 3000 LC system coupled to the Orbitrap QE HF, mobile phase solvents consisted of solvent A (0.1% formic acid, 1% acetonitrile) and solvent B (0.1% formic acid, 89.9% acetonitrile). Peptides were separated using a linear gradient over 25 min, initiated at 3% solvent B and increased to 23% over the first 21 min, then raised further to 38% solvent B over the next 4 min, followed by a wash step at 95% solvent B. The QE HF mass spectrometer was operated in data-dependent acquisition (DDA) mode, automatically switching between MS1 and MS2 scans. Full MS1 spectra were recorded over a mass range of m/z 400–1600 at a resolution of 60,000 at *m/z* 400. MS2 spectra were generated from up to 15 precursor ions using higher-energy collisional dissociation (HCD) with a normalized collision energy (NCE) of 27 and an isolation window of 1.4 *m/z*.

For analyses performed on the Vanquish Neo coupled to the Orbitrap Tribrid Eclipse mass spectrometer, mobile phase solvents consisted of solvent A (0.1% formic acid) and solvent B (0.1% formic acid, 80% acetonitrile). Peptides were separated using a 30 min linear gradient, starting from 4% solvent B and linearly increasing to 32% over 25 min, then further increasing to 49% solvent B within the subsequent 5 min, and finally concluding with a wash step at 99% solvent B. The Orbitrap Tribrid Eclipse was operated in data-dependent acquisition

mode. MS1 spectra were acquired at a resolution of 120,000 with an AGC target of $1.2 \times 10^6$, a maximum injection time (MaxIT) of 50 ms, an RF lens setting of 30%, and a mass range of *m/z* 400–1600. Dynamic exclusion of precursor ions was set to 10 s, with all charge states of a given precursor excluded upon selection. Monoisotopic peak determination was enabled, with the isolation window centered on the most abundant peak. Singly charged ions and ions with charge states above six were excluded, and an intensity threshold filter was set to $5 \times 10^3$. MS2 spectra were acquired in the linear ion trap using a turbo scan rate, custom AGC target set to $2.5 \times 10^4$, MaxIT of 14 ms, and NCE (HCD) set to 30%.

Raw data obtained from the Orbitrap QE HF instrument were processed using Proteome Discoverer 2.5 software with the Sequest HT search algorithm (Thermo Fisher Scientific)[73]. Fragment ion mass tolerance was set to 0.02 Da, and precursor ion mass tolerance was set to 5 ppm. Trypsin was specified as the digestion enzyme. Carbamidomethylation of cysteine residues was set as a fixed modification, whereas oxidation of methionine, deamidation of asparagine and glutamine, acetylation at protein N-terminus, and methionine loss at protein N-terminus were specified as variable modifications. Peptide quantification was conducted using the precursor ion quantifier node employing the Top N Average method, with N set to 3 for protein abundance calculation.

Raw data from the Orbitrap Tribrid Eclipse instrument were processed using Proteome Discoverer 3.1 software with Sequest HT (Thermo Fisher Scientific)[73]. Fragment ion mass tolerance was set to 0.6 Da, and precursor ion mass tolerance to 10 ppm. Trypsin was again selected as the digestion enzyme, carbamidomethylation of cysteine residues as a fixed modification, and oxidation of methionine and methionine loss at the protein N-terminus as variable modifications. Peptide quantification utilized the precursor ion quantifier node, with protein abundance calculated via the Summed Abundance method.

MS/MS spectra were searched against several protein sequence databases, including a customized contaminant database (part of MaxQuant, MPI Martinsried)[74], UniProt *E. coli* protein database (UP000000625_83333; modified November 2019, containing 4,349 sequences), UniProt *A. thaliana* protein database (UP000006548_3702; modified November 2019, containing 27,402 sequences), UniProt *H. sapiens* protein database (modified June 2020, containing 20,531 sequences), UniProt *D. melanogaster* protein database (modified November 2019, containing 13,776 sequences), and a custom database containing modified sequences of the protein of interest.

## Mass photometry

Mass photometry measurements were conducted using a Refeyn TwoMP mass photometer (Refeyn Ltd, Oxford, UK). Videos were recorded for 1 min with the default image size using Refeyn AcquireMP 2024 R1 software (Refeyn Ltd, Oxford, UK). Freshly washed and dried high-precision microscope coverslips (24 ×50 mm) were used for the measurements. A silicone gasket with six cavities was positioned at the center of the coverslip to create measurement wells. For each measurement, 19 µl of MP buffer (20 mM HEPES, pH 7.4, 150 mM KCl, 1 mM MgCl$_2$, 1 mM EGTA or 1xPBS) was added to each gasket cavity, and autofocus was performed before every measurement. Subsequently, 1 µl of protein solution at a concentration of 400 nM was mixed with the MP buffer. Data analysis and plotting were carried out using Refeyn DiscoverMP 2024 R1 software (Refeyn Ltd, Oxford, UK). A standard contrast-to-mass calibration curve was generated using bovine serum albumin (BSA, 66 kDa) and immunoglobulin G (IgG, 150 kDa and 300 kDa).

## Tubulin co-sedimentation assay

Taxol-stabilized MTs were nucleated in vitro based on a previously published protocol[67]. Briefly, purified porcine brain tubulin was mixed

in BRB80 buffer (80 mM PIPES, 1 mM MgCl2, 1 mM EGTA, pH 6.8) at a final concentration of 50 μM. The nucleation reaction was performed at 37 °C for 20 min in the presence of 0.5 mM GTP (Sigma Aldrich) and 5 μM paclitaxel (Thermo Fisher Scientific) in BRB80-MT buffer (80 mM PIPES, 1 mM MgCl$_2$, 1 mM EGTA, 12.5% (w/v) glycerol, pH 6.8). MTs were subsequently layered on top of a 50 μl 30% (w/v) glycerol cushion containing paclitaxel and GTP and pelleted at 200,000 g for 10 min at 27 °C using an S100-AT3 rotor (Thermo Fisher Scientific). The supernatant was discarded, and the pellet was resuspended in 40 μl of BRB80-MT buffer for further experiments. For co-sedimentation assays, MT concentration in each reaction refers to α/β-tubulin heterodimer molarity. Taxol-stabilized MTs were incubated with 1 μM purified N-clamp, 0.25 mg/ml BSA, and BRB80-MT buffer in a final reaction volume of 30 μl. After 20 min of incubation at room temperature to reach binding equilibrium, the reaction mixtures were sedimented at 200,000$g$ for 10 min at 27 °C in an S100-AT3 rotor. Pellet and supernatant fractions were analyzed by SDS-PAGE (12% or 4-20% SDS Precast gel; BIO-RAD) and Coomassie staining (Coomassie Brilliant Blue G250; Sigma Aldrich). Gel images were acquired using an LAS400IR, and gel densitometry was performed with Fiji software[69]. Only samples that did not co-sediment in the negative control (0 μM tubulin) were used for MT binding quantification. For the comparative analysis, only concentration data points present in every sample were quantified. Normalized MT binding data were obtained by integrating peak intensity values for HAUS6/AUG6/Dgt6 bands or the combined HAUS6/HAUS8 bands. Normalized MT binding data were calculated as the fraction of protein co-sedimented with MTs (pellet fraction) and fitted to a modified quadratic equation as previously[40]:

$$\text{complex\_fraction} = 2\left(B_{max} + K_d + x\right) - 2\left(\left(B_{max} + K_d + x\right)^2 - 4xB_{max}\right)^{1/2}$$

where $B_{max}$ is the maximal fraction of N-clamp bound to MTs, $K_d$ is the apparent dissociation constant, and $x$ is the tubulin heterodimer concentration. Fitting and kinetic parameter calculations were performed using least squares fitting with the SciPy Python library[75], and data were plotted using Matplotlib[76].

## AlphaFold2 predictions
AF2 predictions were performed using AlphaFold v2.2.0, 2.3.1 and 2.3.2 in multimer mode with default settings, yielding 25 models per prediction. PAE values were plotted using PAE viewer[77]. A list of AF2 predictions and the identifiers of the used sequences is given in Supplementary Table 7.

## Conservation of HAUS8
Sequences of HAUS8 and its orthologs were fetched from UniProt for the following organisms: *Homo sapiens* (Q9BT25), *Mus musculus* (Q99L00), *Xenopus laevis* (Q0IHJ3), *Danio rerio* (A0A2R8QND7), *Drosophila melanogaster* (Q9W4M8), *Arabidopsis thaliana* (Q9SUH5), *Emericella nidulans* (Q5BGP4), *Acanthamoeba castellanii* (L8HF95). MSA was generated with the MAFFT algorithm[78] using standard parameters and visualized in JalView[79].

## Conservation of CH domains
The conservation scores of human augmin CH domains shown in Fig. 3 were estimated using the ConSurf server[80] on PDB-7SQK with standard parameters. For the multiple sequence alignment (MSA) of HAUS6, sequences of its orthologs were fetched for the following organisms according to their accession numbers: *Homo sapiens* (Q7Z4H7), *Danio rerio* (A0JMF7), *Drosophila melanogaster* (Q9VAP2), *Xenopus laevis* (A0JPI0), *Bos taurus* (E1BG87), *Mus musculus* (Q6NV99), *Canis familiaris* (A0A8C0MNQ7), *Rattus norvegicus* (A0A0G2K4Y6), *Arabidopsis thaliana* (Q94BP7), *Acanthamoeba castellanii* (XP_004336105.1), *Apis mellifera* (NP_001242967.1),

*Aspergillus nidulans* (XP_663791.1), *Oryza sativa* (XP_015625562.1), *Chlamydomonas reinhardtii* (XP_042915056.1), *Physcomitrium patens* (XP_024393130.1), *Selaginella moellendorffii* (XP_024525773.1). MSA was generated with the MAFFT algorithm[78] using standard parameters and sequence identity of the residues was calculated with JalView[79].

## Conservation of HAUS7 and HAUS8 functional elements
Orthologs of HAUS7/Msd5 and HAUS8/Dgt4 were identified as annotated in UniProtKB or using PSI-BLAST using A0A0N0U5X0, Q9W0G6, A0A195DAI7 or Q9W4M8 as query sequence. Identifiers for used sequences are provided in Supplementary Table 8. Sequences were compared by multiple sequence alignment using MAFFT through the EMBL-EBI job dispatcher[81]. The HAUS7/Msd5 positively charged loop was defined as a stretch of amino acids with a net charge of at least +3 aligning with the positively charged loop in *D. melanogaster* Msd5$_{HAUS7}$. Human HAUS8 residue 142 and accordingly aligned residues were defined as the final residue of the HAUS8/Dgt4-NT. HAUS8/Dgt4-NT charge was calculated as $(K + R) - (D + E) + 1$, accounting for the charge of the N-terminal amino group.

## Generating the *D. melanogaster* augmin holocomplex model
The model of *D. melanogaster* augmin octamer was generated following a 'divide-and-conquer' approach adapted from[29]. Briefly, the highest-scoring model predicted for TIII Dgt3/5$_{HAUS3/5}$ in complex with TII was superposed to the highest-scoring model for TIII using the matchmake command in UCSF Chimera (v. 1.16[82] according to the following residues: Dgt3$_{HAUS3}$ 135-173, 370-392 and Dgt5$_{HAUS5}$ 139-169, 395-429. Afterwards, the two models were merged in UCSF Chimera[82] by retaining complementary segments of Dgt3$_{HAUS3}$ and Dgt5$_{HAUS5}$ from the two predicted models. The model predicted for the TIII HAUS$_{Dgt}$3/5 in complex with TII contributed residues 174-369 from Dgt3$_{HAUS3}$ and residues 170-510 from Dgt5$_{HAUS5}$, as well as the TII tetramer. The model predicted for TIII contributed to the remaining segments of Dgt3$_{HAUS3}$ and Dgt5$_{HAUS5}$, as well as Wac$_{HAUS1}$ and Dgt2$_{HAUS4}$. In the final step, atom numbering was adjusted using the pdb_reatom command from pdb-tools[83].

## Negative stain EM and 2D class averaging
Copper/palladium 400 EM mesh grids (G2400D, Plano GmbH) coated with continuous carbon (~10 nm thickness) were glow-discharged before sample application. 5 μl of purified augmin N-clamp sample was applied to the grids and incubated for 30 sec. Grids were then blotted with Whatman filter paper 50 and washed with three drops of distilled water. Sample staining was done using 3% uranyl acetate in water, and excess staining solution was removed by blotting. All datasets were acquired at RT at a Talos L120C TEM (Thermo Fisher Scientific) operated at 120 kV, equipped with a 4k x 4k Ceta CMOS camera (Thermo Fisher Scientific). Micrographs were taken using the EPU software package (version 2.9, Thermo Fisher Scientific) at a nominal defocus of approximately −2 μm and an object pixel size of 0.1992 nm. The final datasets included 303 images of human augmin N-clamp, 245 images of *X. laevis* augmin N-clamp, 549 images of *A. thaliana* augmin N-clamp and 205 images of *D. melanogaster* augmin N-clamp. Image processing was performed in RELION 3.1[84,85]. Contrast transfer function (CTF) parameters were estimated using Gctf[86]. Automated particle picking was done using 2D class averages obtained after manual particle picking. The selected 2D class averages were low-pass filtered to 30 Å and used as templates for all datasets with a mask diameter of 250 Å or 300 Å. The number of automatically picked particles was 50,630 for human augmin N-clamp, 32,928 for *X. laevis* augmin N-clamp, 135,869 for *A. thaliana* augmin N-clamp, and 43,840 for *D. melanogaster* augmin N-clamp. Particles were extracted in 128 or 256-px

boxes, either rescaled (3.984 Å/px) or at full spatial resolution (1.992 Å/px). Extracted particles underwent up to three rounds of 2D classification into 50-200 classes, with a T-factor of 2 and a circular mask of 250 or 300 Å.

### Cryo-EM of MTs preincubated with wild-type augmin N-clamp

After AEC, purified wild-type *D. melanogaster* augmin N-clamp was desalted in BRB80 using a HiTrap 5 ml column followed by concentration using Amicon, 30 kDa MWCO (Merck) to a final concentration of 4 µM. Taxol-stabilized MTs were prepared as described above and diluted to a concentration of 1 mg/ml in BRB80 buffer at RT. Before plunge freezing taxol-stabilized MTs were mixed in a 1:1 ratio with the desalted *D. melanogaster* augmin N-clamp (4 µM concentration). Holey carbon grids (Cu R2/1, 200 mesh; Quantifoil) were glow-discharged twice for 1 min using an easiGlow plasma cleaner (PELCO). 3 µl of the incubation mixture was applied on the carbon side of the EM grids mounted into a Vitrobot Mark IV (Thermo Fisher Scientific) operated at 30 °C and 100% humidity. After 30 s of incubation, 4 µl of purified *D. melanogaster* augmin N-clamp sample (4 µM concentration) was added to maximize binding. After a waiting time of 1 min and 40 s, the grids were blotted for 1 s from both sides and plunge-frozen in liquid ethane. The grids were screened at a Glacios TEM (Thermo Fisher Scientific) to assess their quality and representative images were collected at different magnifications.

### Cryo-EM of MTs decorated with wild-type augmin N-clamp

Purified *D. melanogaster* augmin N-clamp and MTs were prepared as described above (pre-incubation). Holey carbon grids (Cu R2/1, 200 mesh; Quantifoil) were glow-discharged twice for 1 min using an easiGlow plasma cleaner (PELCO). 3 µl of MT solution (1 mg/ml) was applied on the carbon side of the EM grids and MTs were allowed to adsorb for 1 min at RT and ambient atmosphere before a short manual blotting step (Whatman filter paper #1). Subsequently, the grids were incubated three times with 3 µl of purified *D. melanogaster* augmin N-clamp sample (4 µM concentration), waiting 2 min and manually removing excess buffer (Whatman filter paper #1) between each application. Afterwards, the grids were mounted into a Vitrobot Mark IV (Thermo Fisher Scientific) operated at 30 °C and 100% humidity. Finally, 4 µl of purified *D. melanogaster* augmin N-clamp sample (4 µM concentration) was added to maximize binding. After a waiting time of 1 min and 40 s, the grids were blotted for 1 s from both sides and plunge-frozen in liquid ethane. Grids were stored in liquid nitrogen until further use.

### Cryo-EM of MTs decorated with augmin GST-N-clamp

MTs were prepared as described above and diluted to a final concentration of (0.5 mg/ml). Purified *D. melanogaster* augmin GST-N-clamp was desalted and concentrated as described above to a final concentration of 4 µM. Holey carbon grids (Cu R2/1, 200 mesh; Quantifoil) were glow-discharged twice for 1 min using an easiGlow plasma cleaner (PELCO). 3 µl of MT sample (0.5 mg/ml) was applied on the carbon side of the EM grids and MTs were allowed to adsorb for 1 min at RT and ambient atmosphere before a short manual blotting step (Whatman filter paper #1). Subsequently, the grids were incubated four times with 3 µl of purified *D. melanogaster* augmin GST-N-clamp sample (4 µM concentration), waiting 30 s and manually removing excess buffer (Whatman filter paper #1) between each application. Afterwards, the grids were mounted into a Vitrobot Mark IV (Thermo Fisher Scientific) operated at 30 °C and 100% humidity. Finally, 4 µl of purified *D. melanogaster* augmin GST-N-clamp sample (4 µM concentration) was added and after 30 s the grids were blotted for 1 s from both sides and plunge-frozen in liquid ethane. Grids were stored in liquid nitrogen until further use.

### Cryo-EM data acquisition

Micrograph movie stacks were recorded on a Titan Krios G1 (Thermo Fisher Scientific), operated at 300 kV and equipped with a K3 (Gatan, Inc.) direct electron detector and a Quantum Gatan Imaging Filter (Gatan, Inc.) with a slit width of 20 eV. Automated high-resolution data acquisition was conducted in EPU (Thermo Fisher Scientific) with four images per hole at a nominal magnification of ×81,000 (1.069 Å/px) and a nominal defocus ranging from -1.0 µm to −3.0 µm. The 8528 micrographs of *D. melanogaster* augmin N-clamp were acquired with a cumulative dose of 49.8 e⁻/Å² and a dose rate of 12.8 e⁻/Å²/s distributed over 50 fractions. For the augmin GST-N-clamp 8,882 micrographs were acquired with a cumulative dose of 40.4 e⁻/Å² and of 12.4 e⁻/Å²/s. distributed over 50 fractions. Details of data collection are summarized in Supplementary Table 9.

### Processing of WT augmin N-clamp-MT cryo-EM data

Initial steps of cryo-EM data processing were performed using RELION 3.1[84,85]. Beam-induced motion was corrected with MotionCor2[87] using 5×5 patches, and CTF parameters were estimated using Gctf[86]. For particle picking, 25 micrographs were randomly selected to train a filament-picking model in crYOLO[88] (version 1.8). MT segments were then located on the full dataset using the trained crYOLO model, with a spacing of 80 Å along the MT axis. 441,250 MT segments were extracted in 600 px boxes, rescaled to 5.01 Å/px and were subjected to 2D classification. 423,115 retained MT segments were classified by protofilament (PF) number in 4 random subsets through supervised 3D classification, as described previously[46], with 13PF (149,266 segments) and 14PF (192,313 particles) MTs as most abundant classes.

14 PF MTs were used for further processing using the MiRP pipeline (v2)[46]. Following seam correction, 102,166 MT segments were extracted at full spatial resolution (1.069 Å/px, 432 px box). Extracted MT segments were subjected to refinement with C1 symmetry using parameters outlined in the MiRP manual[46], employing a loose mask with a z-length of 30% of the MT and the final 3D class of the MiRP pipeline before seam correction as reference. Afterwards, the outputs of the particle refinement runs were used as references.

From here on, processing was performed in RELION 3.0, unless mentioned otherwise. Successive CTF refinement and Bayesian polishing (each performed twice) interspersed with 3D refinements (C1 symmetry) were applied, followed by an additional round of C1 refinement. From this point, MT segments were processed through two branches. In one branch, MT segments were subjected to helical refinement with a rise of 8.7 Å and a twist of -25.8 Å applying 14-fold helical symmetry, reaching a global resolution of 4.0 Å, followed by sharpening according to the automatically determined B-factor and filtering to local resolution. In the second branch, MT segments were symmetry expanded after the aforementioned C1 refinement using the relion_particle_symmetry_expand command, with a 14-fold expansion using a twist of -25.76 Å and a rise of 8.71 Å, yielding 1,430,324 MT segments. Expanded MT segments were subsequently subjected to C1 refinement using the same parameters as the initial refinement and transferred for further processing to CryoSPARC 4.5.3[89].

In CryoSPARC, MT segments were classified into 3 classes using a soft focus mask covering an α/β-tubulin dimer and associated decorating protein density opposite the seam. The class containing the most well-defined decorating density in uniform tubulin register was selected (469,266 MT segments) and MT segments were subjected to non-uniform homogeneous refinement (C1 symmetry) with local angular sampling, using a solvent mask covering all 14 protofilaments and decorating protein. Finally, the reconstruction, reaching 4.7 Å, was subjected to local resolution and local filter jobs.

### Processing of augmin GST-N-clamp-MT cryo-EM data

Initial steps of cryo-EM data processing were performed as done as for the wild-type data set. For particle picking in crYOLO, the model

trained with the wild-type data was used. For the GST-N-clamp dataset, 348,721 MT segments at 80 Å spacing were extracted in 600 px boxes, rescaled to 5.01 Å/px and subjected to 2D classification. 335,829 retained MT segments were classified by protofilament (PF) number through supervised 3D classification, as described previously[46], with 13PF (175,657 segments) and 14PF (98,204 particles) as the most abundant MT classes.

13 PF MTs were used for further processing using the MiRP pipeline (v2)[46]. Following seam correction, 141,148 MT segments were extracted at full spatial resolution (1.069 Å/px, 432 px box). Extracted MT segments were subjected to refinement with C1 symmetry using parameters outlined in the MiRP manual[46], employing a loose mask with a z-length of 30% of the MT and the final 3D class of the MiRP pipeline before seam correction as reference. Afterwards, the outputs of the particle refinement runs were used as references.

From here on, processing was performed in RELION 3.0, unless mentioned otherwise. Successive CTF refinement and Bayesian polishing (each performed twice) interspersed with 3D refinements (applying helical symmetry) were applied, followed by an additional round of C1 refinement. Afterwards, MT segments were symmetry expanded after the aforementioned C1 refinement using the relion_particle_symmetry_expand command, with a 13-fold expansion using a rise of 9.521 Å and a twist of -27.657 Å, yielding 1,834,924 MT segments. Expanded MT segments were subsequently subjected to C1 refinement using the same parameters as the initial refinement, reaching 3.6 Å, resolution and transferred to CryoSPARC 4.4.1[89] for further processing.

In CryoSPARC, MT segments were classified into 4 classes using a soft focus mask covering decorating protein density opposite the seam. One class containing well-defined decorating density was selected (457,096 MT segments) and MT segments were subjected to non-uniform homogeneous refinement (C1 symmetry) with local angular sampling, using a solvent mask covering 2×3 tubulin monomers and the central decorating density. The reconstruction, reaching 3.9 Å and was filtered locally. Using the pyEM package[90], the particle set was subsequently transferred back to RELION 3.1, and particles were recentered on the augmin N-clamp-MT interface in 256-pixel box at full spatial resolution. To retain the particles in a polished state, they were recentered using a signal subtraction job without actual signal subtraction, by providing a wide mask around the MT[91]. These particles were reconstructed to generate a 3D reference and subsequently, a 3D refinement was performed with a mask covering 2 × 5 tubulins and the central decorating density. The resulting map was then submitted to 3D classification without alignment in RELION 5.0, with Blush regularization applied[92]. The class with most well-defined N-clamp density was selected, containing 117,870 particles. Finally, the particles were subjected to local refinement and filtered to local resolution. Details of data processing are summarized in Supplementary Table 9.

## Analysis of cryo-EM reconstructions

All rigid-body docking was performed using the fitmap command in ChimeraX (1.3, 1.7.1 and 1.9)[93]. The AF2 prediction of *D. melanogaster* N-clamp was rigid-body docked into the density obtained after 3D classification in CryoSPARC, which had sufficient resolution to distinguish a- and β-tubulin (Supplementary Fig. 10b). For the final cryo-EM density (Supplementary Fig. 10c), an atomic model for one protofilament of a Taxol-stabilized MT (PDB-6WVR) was rigid-body docked twice into adjacent PFs of the density, along with the AF2 prediction of *D. melanogaster* augmin N-clamp (CH6 K10-L145). Then, individual chains from the AF2 prediction of porcine α/β-tubulin (Supplementary Table 7) were separately aligned to the rigid-body docked bovine MT (PDB-6WVR) by matchmaking in ChimeraX. The full *D. melanogaster* N-clamp AF2 prediction was placed into the density by matchmaking its Dgt6$_{HAUS6}$ CH domain onto the Dgt6$_{HAUS6}$ CH domain fitted as described above.

## Molecular dynamics simulations

MD simulations were conducted at atomic-level resolution, using the CHARMM36m force field (February 2021) for proteins, the CHARMM TIP3P force field for water, and the standard CHARMM36 force field for ions[94]. The parameters for GDP and GTP molecules were generated using CHARMM-GUI v1.7, specifically through the Ligand Reader & Modeler tool[95].

All simulations were performed using the GROMACS 2024.2 simulation package[96]. AlphaFold 2.3.2, with default settings, was employed to generate the full-length structures of the α-tubulin (P81947) and β-tubulin (Q6B856) subunits. The predicted structures were aligned using the ChimeraX Matchmaker tool[93] to the tubulin dimer resolved in PDB 6WVR[97], allowing verification of AlphaFold's predictions for the resolved regions. The dimers in the PDB structure contained GDP and GTP molecules as well as magnesium ions, which were correctly incorporated into the AF2-predicted structures.

To model the MT-bound *D. melanogaster* augmin N-clamp complex, we utilized a structure comprising four protofilaments, each containing two α-tubulin and two β-tubulin subunits. The augmin N-clamp was placed based on a rigid body fit into the cryo-EM reconstruction of wild-type N-clamp. N- and C-termini of all proteins were modeled as charged residues due to their location in flexible loops exposed to solvent. Additionally, the proximity of N-terminal groups of both the N-clamp and tubulin dimers predicted by AF2 hypothesized that they could contribute to stabilizing or destabilizing the interaction interface in their charged forms. To retain the overall shape of the truncated MT filament without simulating the entire tube, we constructed a system comprising four rows of tubulin dimers. Overall, the system consisted of 8 α-tubulin and 8 β-tubulin molecules, the *D. melanogaster* augmin N-clamp complex, as well as 8 GDP and GTP molecules, and 8 magnesium ions bound to the tubulins. It also included 298,034 water molecules, 1,026 potassium ions, and 592 chloride ions, totaling 1,016,448 atoms. The system was enclosed in a rectangular simulation box with dimensions 22.97 × 27.07 × 16.57 nm. During the production runs, positional restraints were applied only to the two outer protofilaments, while the two central rows (those directly interacting with the N-clamp) remained unrestrained. To restrain the two outer protofilaments, a force constant of 1000 kJ/mol/nm² was applied in all dimensions to both the side chains and backbone atoms. The periodic boundary conditions were applied in all dimensions and the box size was configured to simulate an infinite MT filament in the z-axis, effectively mimicking a realistic section of a MT filament.

Initially, all systems were energy minimized in a vacuum, followed by hydration. To mimic experimental conditions, we added 150 mM potassium chloride and an appropriate number of counter ions to neutralize the extra charges. Subsequently, an equilibration phase was conducted under NpT ensemble conditions. During this phase, all proteins were restrained with force constants of 400 kJ/mol/nm² for backbone atoms and 40 kJ/mol/nm² for side chains. The V-rescale thermostat maintained the temperature at 303.15 K with a 1.0 ps time constant[98]. A pressure of 1 atm was regulated using the C-rescale barostat with a time constant of 5.0 ps and an isothermal compressibility of 4.5×10⁻⁵ bar⁻¹ [99] and an isotropic pressure-coupling scheme. Neighbor searching utilized the Verlet scheme, updating every 20 steps[100]. Electrostatic interactions were calculated using the particle mesh Ewald method[101] with a 0.12 nm grid spacing, a 10⁻⁵ tolerance, and a 1.2 nm cutoff. Three independent simulations were performed by varying the initial atomic velocities. Each simulation was carried out with a 2 fs time step and extended to a total duration of 1000 ns.

Analyses were conducted using an in-house tool in combination with MDAnalysis[102] for contact, RMSD and RMSF calculations, and standard GROMACS tool (gmx energy) for interaction energy calculations. Interaction energy calculations were used to refine the contact map and distinguish truly interacting residues from those simply

nearby, as previously done in refs. 103,104. Visualization snapshots and video were generated using VMD[105].

## Statistical analysis

Protein band intensities on immunoblots or Coomassie blue-stained gels were quantified using Fiji software (v2.14.0)[69]. Signal intensity measurements using the line plot tool or distance measurements of immunofluorescence images were performed on maximum intensity-projected images using Fiji software[69].

Data plotting and statistical analysis were done using GraphPad Prism (Version 10.0). Data normality was assessed using the Shapiro–Wilk test. Subsequently, data were analyzed using a two-tailed $t$ test, Mann–Whitney $U$ test or the Kruskal–Wallis test with Dunn's multiple comparison post-hoc analysis. A significance threshold of $p < 0.05$ was applied.

## Reporting summary

Further information on research design is available in the Nature Portfolio Reporting Summary linked to this article.

## Data availability

Atomic coordinates and cryo-EM densities have been deposited at the Protein Data Bank and the Electron Microscopy Data Bank under accession codes: PDB 9RPD and EMD-54161 (*D. melanogaster* augmin GST-N-clamp bound to a MT, well-defined subset of particles), EMD-54174 (*D. melanogaster* augmin N-clamp bound to a MT), EMD-52832 (helical reconstruction of a *D. melanogaster* N-clamp-decorated MT), EMD-54160 (*D. melanogaster* augmin GST-N-clamp bound to a MT). Published structural data used in this article are: PDB 7SQK, PDB 6WVR, PDB 3IZ0, PDB 7PT5 and PDB 8AT3. The models predicted by AF2 generated in this study have been deposited in the ModelArchive database with the identifiers ma-vxlav [https://modelarchive.org/doi/10.5452/ma-vxlav] (*X. laevis* augmin N-clamp), ma-apjck [https://modelarchive.org/doi/10.5452/ma-apjck] (human augmin N-clamp), ma-a7krk [https://modelarchive.org/doi/10.5452/ma-a7krk] (*D. melanogaster* augmin N-clamp), ma-u3pr2 [https://modelarchive.org/doi/10.5452/ma-u3pr2] (*A. thaliana* augmin N-clamp), ma-t3gx3 [https://modelarchive.org/doi/10.5452/ma-t3gx3] (*A. thaliana* augmin TII), ma-5p38z [https://modelarchive.org/doi/10.5452/ma-5p38z] (*D. melanogaster* augmin TII), ma-8pbts [https://modelarchive.org/doi/10.5452/ma-8pbts] (*D. melanogaster* augmin TIII), ma-7w5lz [https://modelarchive.org/doi/10.5452/ma-7w5lz] (D. melanogaster augmin TII + III(interface)), ma-3drto [https://modelarchive.org/doi/10.5452/ma-3drto] (*D. melanogaster* augmin CH6 with tubulin dimer), ma-5rtzo [https://modelarchive.org/doi/10.5452/ma-5rtzo] (human augmin CH6 with tubulin dimer), ma-xcax9 [https://modelarchive.org/doi/10.5452/ma-xcax9] (*A. thaliana* augmin CH6 with tubulin dimer), ma-upjzh [https://modelarchive.org/doi/10.5452/ma-upjzh] (*X. laevis* augmin CH6 with tubulin dimer). Mass spectrometry data of purified augmin N-clamp complexes are available from the ProteomeXchange Consortium[106] via the PRIDE partner repository[107] with the dataset identifier PXD060099. The initial configuration files for the Molecular Dynamics simulations associated with this study and the trajectories have been deposited on ZENODO. They are publicly available at the following address: https://doi.org/10.5281/zenodo.16452427[108]. The script used for quantitative analysis and plotting of the tubulin co-sedimentation assays is available at https://github.com/gt-biomodel/tubulin-co-sedimentation-plots and at https://doi.org/10.5281/zenodo.16452427[108]. Source Data are provided as a Source Data file. Source data are provided with this paper.

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

## Acknowledgements

The authors thank Ursula Jäkle, Annika Hanraths, Nethaji Kuruppu and Ramiro Monge Lozano for help in protein expression and purification. We thank Oliver Gruss, Simone Reber, Luca Troman and Mamata Bangera for discussion and support during cryo-EM data processing, and

Anne-Laure Pauleau for technical suggestions. Human *HAUS2, HAUS6, HAUS7* and *HAUS8* genes were a gift from Laurence Pelletier. We thank Jens Lüders for the HAUS6 antibody. We further thank Karine Lapouge from the European Molecular Biology Laboratory (EMBL Heidelberg, Germany) Protein Expression and Purification Core Facility (PEPCF) for Mass Photometry. We acknowledge the technical support of the Core Facility for Mass Spectrometry and Proteomics of the Center for Molecular Biology of Heidelberg University (ZMBH). We thank Marcin Luzarowski and Sabine Merker for support with data analysis and deposition. The Core Facility for Mass Spectrometry and Proteomics is funded by the ZMBH and partially funded by the CellNetworks Core Technology Platform (CCTP) of Heidelberg University. The CCTP is funded in part by the Federal Ministry of Education and Research (BMBF) and the Ministry of Science Baden Württemberg within the framework of the Excellence Strategy of the Federal and State Governments of Germany. The purchase of the Orbitrap Tribrid Eclipse used in this study was funded in part by the German Research Foundation (DFG)—project number: 538758380. Moreover, we would like to acknowledge access to the infrastructure and support provided by the Cryo-EM Network at the Heidelberg University (HDcryoNet), which is funded and supported by the German Research Foundation (DFG), the Federal Ministry of Education and Research (BMBF) and the Ministry of Science Baden-Württemberg, among others, within the framework of the Excellence Strategy of the Federal and State Governments of Germany. We also acknowledge the services SDS@hd and bwHPC supported by the Ministry of Science, Research and the Arts; DFG Sch Baden-Württemberg, as well as the German Research Foundation (INST 35/1314-1 FUGG, INST 35/1503-1 FUGG and INST 35/1134-1 FUGG). We thank EMBL IT for computational and data storage support. This work is supported by grants of the Deutsche Forschungsgemeinschaft (DFG) to ES (DFG Schi 295/4-4; DFG SCHI 295/11-1; DFG SCHI 295/9-1 and to SP (DFG PF 963/1-4; DFG PF 963/4-1), to FL (SFB-1638/1-511488495-Z01, SFB/TRR 186, project A1, DFG LO 2821/1-1) and to MK (SFB-1638/1-511488495-Z01). FL acknowledges the computing resources provided by the CSC-IT Center for Science Ltd. (Espoo, Finland) and supported by the state of Baden-Württemberg through bwHPC and the German Research Foundation (DFG) through grant INST 35/1597-1 FUGG. S.P. also acknowledges funding by the Aventis Foundation and the Chica and Heinz Schaller Foundation. M.W. acknowledges funding by the Health + Life Science Alliance Heidelberg Mannheim.

## Author contributions

M.W., G.T., and M.Z. performed cloning, protein expression, protein purification and tubulin co-sedimentation assays. G.T., M.W., and A.N. performed plunge freezing experiments. G.T., B.J.A.V., and M.H. acquired cryo-EM data. G.T., B.J.A.V., and M.W. analyzed cryo-EM data. A.N. performed negative staining and negative stain EM image acquisition and G.T. and M.W. analyzed negative stain EM data. A.S. and S.E. performed experiments in S2 cells. Q.G. and M.Z. performed microscopy and experiments in RPE1 cells. M.K. and F.L. performed and analyzed MD simulations. G.T., B.J.A.V., and W.M. performed AlphaFold2 predictions. M.W., E.S., S. Eustermann and S.P. supervised the experiments. M.W., G.T., B.J.A.V., E.S., and S.P. wrote the manuscript with input from the other authors. All authors discussed the data and gave final approval for publication.

## Funding

## Competing interests

The authors declare no competing interests.
