## [Transparent Peer Review file · Nature Communications]

Conserved function of the HAUS6 calponin homology domain in anchoring augmin for microtubule branching

Corresponding Author: Dr Stefan Pfeffer

Version 0:

Reviewer comments:

Reviewer #1

(Remarks to the Author)

The augmin complex is a structurally conserved hetero-octamer responsible for recruiting the main MT nucleating complex, g-TuRC, to pre-existing MTs, facilitating mitotic spindle formation and, through this, chromosome segregation and euploidy. In vivo, in vitro and in silico work has previously suggested that the recruitment of Augmin to pre-existing MTs requires the HAUS8 subunit – with recent research focusing on an unstructured/unresolved N terminal region of this protein, alongside regions of two other subunits, which together form the “N clamp”

In this paper, the authors:

Undertake a comparative alphafold2 predicted structural analysis of the T-II (4 subunit) augmin module across a range of model organisms, providing evidence that Drosophila N-clamp contains only 1 CH domain

Recombinantly expressed and purified human, Xenopus, Arabidopsis and Drosophila “N-clamp” (regions of these 4 subunits), undertaking negative stain EM to obtain structural information and confirming their capacity to bind in vitro generated MTs

Repeated the MT binding using purified N-clamp complexes lacking the N-terminal section of HAUS8 homologues. Confirmed previous results that this is essential for MT binding in humans. Demonstrated some concentration-dependent MT binding in Xenopus and Arabidopsis but that Drosophila N-clamp lacking the N-term of HAUS8/Dgt4 still efficiently bound MTs.

Modelled structures of the CH6 and CH7 domains to identify residues likely to be involved in MT binding. Then designed a mutant version of human CH6 in which 4 basic residues in HAUS6 were mutated to Alanine to test, demonstrating reduced ability to bind MTs. When repeated with a Drosophila CH6 mutant version in which 3 corresponding basic residues in Dgt6 were mutated, they found MT binding severely impaired. Co-RNAi/replacement expressions in S2 (Drosophila) cells confirmed reduced spindle localisation of this mutant, in relation to wild type.

Took purified Drosophila N-clamp and purified stabilised porcine MTs and undertook cryo-EM and reconstruction to obtain density structure maps that were fitted, first to the CH6 domain and then to the full Drosophila Augmin predicted structure

Undertook MD simulations to predict the flexibility of the interaction between Drosophila N-clamp and the ab tubulin interface, demonstrating stable interactions. This allows a number of predictions about which specific residues in Msd5 interact with the b-tubulin.

Used AF2 to map the interactions of human, Xenopus and Arabidopsis CH6 with an ab tubulin dimer, finding that similar regions to the Drosophila proteins were top-ranked and generating predicted structures. They then undertook Co-RNAi/replacement expressions in RPE1 cells confirmed reduced spindle localisation of the 4A mutant (as described in the in vitro MT binding assays), in relation to wild type.

The work is original, important and significant. Research to understand, at the molecular level, how Augmin functions has

been slow, mainly due to the complexity of the protein complex. By taking a comparative approach, combining AF2, biochemistry, cryo-EM and cell biology the authors make a strong argument for having identified the precise way in which Augmin binds to MTs.

I do not think additional evidence is needed to support the claims, but I do think that some additional consideration needs to be added to assist the reader in interpreting the cryo-EM data, in relation to the modelling in relation to Figure 4. In particular, Line 243 states – “the additional density clearly recapitulated the CH6 domain shape as predicted by AF2.” This may be my eye but it’s not clear to me that the CH6 domain shape is the only reasonable structure that could provide that density. Indeed, in Figure 4e, the cryoEM density seems not to match the predicted region of the N-clamp next to the CH6 domain. At this point of the paper reader of the paper has not seen the MD simulations (which back up their hypothesis). Can the authors provide a clearer explanatory text to help guide the reader? Or reword in less certain terms at this point of the text?

I would also like to see the authors, in the Discussion, more fully consider why the 4A mutant of HAUS6 has such a dramatic “augmin-like” phenotype in RPE1 cells, yet HAUS8-NT plays the dominant MT-binding role in vitro and in vivo. They do postulate that HAUS-8 may recruit and stabilise the CH6-MT interaction, but this does not explain how *Drosophila* Augmin (which lacks HAUS-8 MT binding activity) can so efficiently bind MTs in vivo. Also, some consideration should be made to the potential difference between Augmin that has been expressed and purified in bacteria and the in vivo complex, which is known to be extensively post-translationally modified during mitosis, when it is most active.

The experiments have been performed to a very high standard, with 3 replicates of experimental techniques, robust quantification and statistical analyses, meeting - and in some cases - exceeding standards in the field.

The one method that needs clarification relates to the spotting of *Drosophila* N-clamp and MTs onto Cryo-EM grids. Line 806-811: The method described is unclear. Does it mean that the 9 μ M tubulin was diluted in equal volume to 4 μ M N-clamp, resulting in a mixture containing 4.5 μ M tubulin heterodimer and 2 μ M N-clamp; to which an additional 4 μ l of 4 μ M N-clamp was added? This should probably be re-written to assist the reader.

Additional minor comments:

Line 218 – remove the word “essential”. The authors write that the CH6 domain is “a conserved and essential MT-binding element”. Yet, it is clear from the data (Figure 3c) – and is stated in the text - that, in the absence of CH6 in humans, the N-clamp still binds MTs; just a lower fraction.

Supplementary Figure 9 – in (d), Dtg2 should be Dgt2

Reviewer #2

(Remarks to the Author)

This manuscript focuses on binding mechanism of the microtubule-interacting region of the augmin complex known as the N-clamp and associated implications for microtubule branching. Using structural prediction and co-sedimentation assays, this work identifies the minimal microtubule-binding region of augmin across four species to be the HAUS6 CH domain, identifying more or less contribution to binding of accessory regions, in particular the disordered HAUS8 N-terminus. The *Drosophila* N-clamp was further investigated with mutational and structural analysis, as it lacks the HAUS7 CH and has a particularly short HAUS8 N-terminus, such that HAUS6 CH is the key mediator of microtubule interaction. Alphafold predictions combined with cryo-EM and molecular dynamics analysis suggest a model for HAUS6 CH interaction with the interprotofilament groove between beta-tubulins. Finally, the HAUS6 CH was shown to be of likely importance in the binding of human augmin to microtubules, based on mutagenesis studies on mitotic spindle architecture in cells.

A key finding of this work is the importance of the HAUS6 CH domain in augmin microtubule interaction across species and the variable contribution of the HAUS8 N-terminus, which was previously thought to be more central to the interaction (although it is required in humans).

While the non-structural parts of the study are sound, my main issue with this work is that the cryo-EM analysis is not of sufficient quality to unambiguously define the binding mode of the Dm HAUS6 CH. Without sufficient improvement in the quality of cryo-EM density for the HAUS6 CH, the study is too reliant on Alphafold predictions and not suitable for publication. Obviously, the CH density being at near-atomic resolution would be the ideal situation, but this is not always possible. However, given the dataset size and the lack of flexibility in the CH indicated by Alphafold and molecular dynamics, I don’t see why this density should not be at a higher resolution. With a good sample (including good decorating protein occupancy) this microtubule processing pipeline should be producing higher resolution details at least with the version operating in Relion v3.0 (for example see Cook et al., 2020, or Atherton et al., bioRxiv 2024 (<https://doi.org/10.1101/2024.11.11.622991>)).

The CH required low-pass filtering, which makes unambiguous fitting (even rigid-body) challenging for a small globular domain. Indeed, looking at the provided reconstructions, the alpha-beta tubulin register is not particularly well resolved and the density for the CH domain only appears at low threshold. The fit is not great even at this resolution, particularly for the two longest helices in the CH domain. This raises the possibility of some inaccuracy in the alphafold predictions, or a

conformational change upon microtubule association that has not been modelled. My feeling from the data provided and considering the concentrations of N-clamp used is that sub-stoichiometric occupancy of the N-clamp on the microtubule may be an issue in this dataset and/or suboptimal image processing. If the authors can improve the cryo-EM reconstructions/model in a revised submission, it would be suitable for publication.

Other specific comments;

- 1) Sup Fig 3 c is a little unconvincing. A higher mag view of the sample would be more useful.
- 2) Line 145 should also refer to Figs showing co-sedimentation raw data examples.
- 3) Line 227 is worded in such a way it seems there may have been some augmin-mediated nucleation of MTs from pre-existing taxol-stabilised MTs. This is not the case and this should be made more clear.
- 4) Line 228- could the authors comment on whether the bundling represents any feature of biological interest?
- 5) Line 237. EM analysis. The local resolution shown indicates that the resolution of the N-clamp is ~ 6-10Å. Why then is a 15Å low pass filter applied?
- 6) Figures 4b/c should be shown with taxol model for full assessment of resolving tubulin register.
- 7) Line 253. In Fig sup 7e we can see some of the rest of the N-clamp in the 2D classes. This makes it strange that we can't see much in 3D. Why might this be? Does it appear at different thresholds or with different filtering?
- 8) Line 303. I'd dispute the high similarity- there is quite a bit of the model outside the density!
- 9) Cryo-EM methods section- please indicate final concentrations of N-clamp used in sample prep more clearly.
- 10) What was the reference used during cryo-EM microtubule filament refinements? This is very important for understanding what has happened during processing.

Reviewer #3

(Remarks to the Author)

I am a specialist in atomistic molecular dynamics simulations, and I have also used these in conjunction with cryo-EM data. While I do have an interest in molecular motors and microtubules, this is not my main area of expertise. I am not an expert in the experimental protocols used in the paper so I will not comment on their validity. I will also leave it to the experts on the cytoskeleton to comment on whether this publication introduces sufficiently new information compared to Zupa et al Nat Comms 2022 to be novel enough to merit another Nat Comms paper, however, from my perspective the additional atomistic detail about the binding interface does merit publication in a high impact journal.

The use of alpha-fold models to build the initial structures for fitting into the cryo-EM density is very sensible, and it is interesting to compare across different species. I am presuming the authors chose the *Drosophila* structure because it does not contain as many disordered regions in the alpha-fold model as the other species they modelled. Please can the authors clarify this in the paper.

I also became slightly confused about the E-hook and the missing loop region in *Drosophila* relative to the other species and whether it is present or absent in the structure/sequence. I read the text as meaning it is absent from both structure and sequence in the fly protein, but in Fig 1b it does look like there could be missing residues in the circled area (but clearly it is not possible for the reader to know this for certain without checking the sequence).

In their MD methods, please can the authors state whether the termini of protein chains are charged or neutralised with capping groups in their simulations, and why explain why this decision was made. Could they also make it clearer whether the production run MD is unrestrained, or whether they have kept restraints on the backbone and/or sidechains. Could they also please explain why only the final 200ns was used in the analysis and why this segment was chosen - for example did they see significant structural relaxation in the first 800ns of the trajectory? Please could they also explain what is measured by the function `gmx_energy` and justify the use of this measurement in their analysis. Energy calculations that involve electrostatics are subtle in MD simulations, because the presence of the solvent screens the charges, so often implicit Poisson-Boltzmann models (or approximations thereof) are used.

Have the authors considered sharing a pdb file containing the structures used to run their simulations, as well as the truncated structure (model.cif) in their zip file containing the cryo-EM maps and models? What differences (if any) did the authors notice between their independent repeat simulations?

Overall I would assess the simulations that have been run as high quality, given that there are 3 independent repeats, and that the simulation timescales correspond to current state of the art (although the 200ns segment used to perform the analysis is somewhat shorter than usual). I would have liked to be able to look at the structures sampled during the MD, however the movie provided as supplementary information was useful.

Version 1:

Reviewer comments:

Reviewer #1

(Remarks to the Author)

I am pleased that the authors have comprehensively addressed all my comments, and have undertaken additional CryoEM

studies that enhance their conclusions. This further strengthens the manuscript and I'm happy to recommend it for publication.

Reviewer #2

[Editorial Note: Reviewer #2's attachments are displayed at the end of this file]

(Remarks to the Author)

Lots of hard work has been done to improve the reconstruction via a combination of engineering a dimer for better affinity/occupancy and improvements in the image processing methods. The rigid fitting now looks way more convincing for the N-clamp, particularly the CH domain Haus6 region. The authors have addressed my main concern here, but I would like to make two suggestions- 1) is to show the same model in the Figures as has been provided. For example, Haus8's fitted helix is longer in figure 4 than it is in the deposited model (see attached images of figure 4 vs a chimera session with the model/map). The angle of the image suggests a good fit, but I expect this is an exaggeration. 2) I would say the helices in the non-CH domain region of the N-clamp (Haus 8/7 and one from HAUS6) could arguably be extended and flexibly fitted into the map- it seems like their orientation may be different to the alphafold model and this could be interesting for functional interpretation. It could also just be that the map is poorer/lower resolution in this region because of flexibility- which fits with the MD simulations- hindering a good fit. I do understand the hesitance to flexibly fit even at this resolution- so I would suggest that the authors just mention/discuss the not-so-good fit of these regions relative to the core of the CH domain.

Reviewer #3

(Remarks to the Author)

The authors have carefully responded to their comments from the referees, and I am satisfied that the simulations described in the manuscript are of high quality. I would suggest trying more sophisticated energy calculations (e.g. MMPBSA) when assessing energy decompositions in the future, to see if that provides any additional insight. I presume that the simulations are the same as previously and have not been rerun following the improved structure determination - please can the authors justify this decision in their subsequent response to reassure the editor that this is appropriate.

REVIEWER COMMENTS

Reviewer #1 (Remarks to the Author):

The augmin complex is a structurally conserved hetero-octamer responsible for recruiting the main MT nucleating complex, γ -TuRC, to pre-existing MTs, facilitating mitotic spindle formation and, through this, chromosome segregation and euploidy. In vivo, in vitro and in silico work has previously suggested that the recruitment of Augmin to pre-existing MTs requires the HAUS8 subunit – with recent research focusing on an unstructured/unresolved N terminal region of this protein, alongside regions of two other subunits, which together form the “N clamp”

In this paper, the authors:

Undertake a comparative alphafold2 predicted structural analysis of the T-II (4 subunit) augmin module across a range of model organisms, providing evidence that Drosophila N-clamp contains only 1 CH domain. Recombinantly expressed and purified human, Xenopus, Arabidopsis and Drosophila “N-clamp” (regions of these 4 subunits), undertaking negative stain EM to obtain structural information and confirming their capacity to bind in vitro generated MTs

Repeated the MT binding using purified N-clamp complexes lacking the N-terminal section of HAUS8 homologues. Confirmed previous results that this is essential for MT binding in humans. Demonstrated some concentration-dependent MT binding in Xenopus and Arabidopsis but that Drosophila N-clamp lacking the N-term of HAUS8/Dgt4 still efficiently bound MTs.

Modelled structures of the CH6 and CH7 domains to identify residues likely to be involved in MT binding. Then designed a mutant version of human CH6 in which 4 basic residues in HAUS6 were mutated to Alanine to test, demonstrating reduced ability to bind MTs. When repeated with a Drosophila CH6 mutant version in which 3 corresponding basic residues in Dgt6 were mutated, they found MT binding severely impaired. Co-RNAi/replacement expressions in S2 (Drosophila) cells confirmed reduced spindle localisation of this mutant, in relation to wild type.

Took purified Drosophila N-clamp and purified stabilised porcine MTs and undertook cryo-EM and reconstruction to obtain density structure maps that were fitted, first to the CH6 domain and then to the full Drosophila Augmin predicted structure. Undertook MD simulations to predict the flexibility of the interaction between Drosophila N-clamp and the α tubulin interface, demonstrating stable interactions. This allows a number of predictions about which specific residues in Msd5 interact with the β -tubulin.

Used AF2 to map the interactions of human, Xenopus and Arabidopsis CH6 with an $\alpha\beta$ tubulin dimer, finding that similar regions to the Drosophila proteins were top-ranked and generating predicted structures. They then undertook Co-RNAi/replacement expressions in RPE1 cells confirmed reduced spindle localisation of the 4A mutant (as described in the in vitro MT binding assays), in relation to wild type.

#1 The work is original, important and significant. Research to understand, at the molecular level, how Augmin functions has been slow, mainly due to the complexity of the protein complex. By taking a comparative approach, combining AF2, biochemistry, cryo-EM and cell biology the authors make a strong argument for

having identified the precise way in which Augmin binds to MTs.

We thank the reviewer for the positive assessment of our integrative approach to dissect the augmin-MT interaction.

#2 I do not think additional evidence is needed to support the claims, but I do think that some additional consideration needs to be added to assist the reader in interpreting the cryo-EM data, in relation to the modelling in relation to Figure 4.

To address this comment and the comments of Reviewer #2 related to cryo-EM structural analysis, we performed extensive additional cryo-EM experiments, including structural analysis of a dimerized N-clamp construct with strongly improved MT binding affinity. This enabled us to generate a cryo-EM reconstruction of CH6 with resolved secondary structure elements, allowing for highly confident and unambiguous docking of the AF2-predicted model onto microtubules. Please refer to our response to the comments of Reviewer #2.

#3 In particular, Line 243 states – “the additional density clearly recapitulated the CH6 domain shape as predicted by AF2.” This may be my eye but it’s not clear to me that the CH6 domain shape is the only reasonable structure that could provide that density.

Using the GST-N-clamp construct with strongly improved MT binding affinity provided a cryo-EM reconstruction in which secondary structure elements of CH6 were clearly resolved, confidently identifying the decorating density as the *D. melanogaster* N-clamp. Please refer to our response to comments from Reviewer #2 for more details.

#4 Indeed, in Figure 4e, the cryoEM density seems not to match the predicted region of the N-clamp next to the CH6 domain. At this point of the paper reader of the paper has not seen the MD simulations (which back up their hypothesis). Can the authors provide a clearer explanatory text to help guide the reader? Or reword in less certain terms at this point of the text?

In our improved cryo-EM reconstruction of the GST N-clamp construct, more distal regions of the augmin N-clamp are now resolved as well. However, density for these more distal segments was lower resolution and more fragmented as compared to the CH6 domain, indicating higher flexibility, consistent with our MD simulations. Because cryo-EM is an averaging based approach, the signal for more flexible protein segments is averaged out and those segments are less well represented in the reconstruction.

In response to this comment, we furthermore moved our integrated model for the positioning of the augmin holocomplex on the MT lattice, such that it is shown only after the MD simulations have been introduced.

#4 I would also like to see the authors, in the Discussion, more fully consider why the 4A mutant of HAUS6 has such a dramatic “augmin-like” phenotype in RPE1 cells, yet HAUS8-NT plays the dominant MT-binding role in vitro and in vivo. They do postulate that HAUS-8 may recruit and stabilise the CH6-MT interaction, but this does not

explain how *Drosophila* Augmin (which lacks HAUS-8 MT binding activity) can so efficiently bind MTs *in vivo*.

Our data indicate that the HAUS8-NT flexibly tethers augmin to MTs, before the extended interface between CH6 and beta-tubulins rigidly anchors and orients augmin on the MT lattice, which is most likely essential for directed nucleation of branched MTs in a pre-defined angle with respect to the pre-existing MT. To clarify this, we extended the discussion section:

“While the unordered HAUS8-NT is unlikely to provide a uniform and stable platform for orienting the γ -TuRC on pre-existing MTs, the conserved CH6 domain and its defined binding site on the MT surface is a prime candidate to fulfill this function. Consistently, the CH6-4A mutations in the human augmin N-clamp only marginally reduced augmin-MT binding in pelleting assays due to the contribution of HAUS8-NT, but functional analyses showed severely disrupted spindle function and assembly in cells expressing HAUS64^{A-mut}, as witnessed by reduced pole-to-pole distance, a phenotype that is typical for loss of augmin function^{13,14}.”

Moreover, motivated by this comment, we aimed to understand which structural elements could take over the role of the HAUS8-NT in *D. melanogaster*. Our MD simulations suggested that a fly-specific loop region of Msd5^{HAUS7} may substantially contribute to the N-clamp-MT interaction in *D. melanogaster* by providing electrostatic interactions reminiscent of the vertebrate and plant HAUS8-NT. To investigate this hypothesis, we generated N-clamp deletion constructs lacking the Msd5^{HAUS7} loop and investigated MT binding *in vitro*. We found that in agreement with our MD simulation data, this loop was indeed important for MT binding, suggesting it could serve as a functional replacement for the HAUS8-NT in other eukaryotes. Overall, this supports our interpretation for the impact of the 4A CH6 mutant in RPE1 cells and further suggests that stable anchoring of CH6, which orients augmin on MTs, generally may require additional electrostatic interactions, provided for instance by the positively charged human HAUS8-NT or the *D. melanogaster* Msd5^{HAUS7} loop.

These new data are shown in Supplementary Fig. 14 of the revised manuscript.

In addition, we have extended the discussion section on the HAUS8-NT and functional replacement by the Msd5^{HAUS7} loop:

“In case of *D. melanogaster*, our MD simulations and tubulin co-sedimentation assays for instance indicate that the very short Dgt4^{HAUS8-NT} could be partially replaced by the disordered loop of Msd5^{HAUS7}, which engaged tubulin E-hook residues through a patch of positively charged residues (K86, K87, R88, R92) in the simulations — reminiscent of the K-loop of KIF1A^{47,51}. Overall, this suggests that stable anchoring of CH6 generally may require additional electrostatic interactions, provided for instance by the positively charged human HAUS8-NT or the *D. melanogaster* Msd5^{HAUS7} loop. In some instances, posttranslational modifications such as phosphorylation may fine-tune those electrostatic interactions, as reported for vertebrate and plant HAUS8-NT⁵²⁻⁵⁴.”

#5 Also, some consideration should be made to the potential difference between Augmin that has been expressed and purified in bacteria and the *in vivo* complex, which is known to be extensively post-translationally modified during mitosis, when it

is most active.

We thank the reviewer for this important remark. We emphasize this aspect now in the results section, when summarizing the functional properties of the augmin N-clamp constructs used:

“Notably, the bacterial expression system does not replicate the extensive post-translational modifications that augmin subunits undergo during mitosis. As a result, the properties of the augmin N-clamp observed in biochemical assays with recombinant augmin N-clamp may represent only a subset of its properties *in vivo*. However, our results demonstrate that the augmin N-clamp is a highly suitable, low-complexity model for dissecting the interaction between augmin and MTs *in vitro* from both mechanistic and structural perspectives.”

#6 The experiments have been performed to a very high standard, with 3 replicates of experimental techniques, robust quantification and statistical analyses, meeting - and in some cases - exceeding standards in the field.

We thank the reviewer for this comment. We have further included additional repetitions of the *in vivo* experiments in *D. melanogaster* S2 cells (Fig. 3) and included a quantification of the three replicates of siRNA controls from RPE1 cells (Fig. 6).

#7 The one method that needs clarification relates to the spotting of Drosophila N-clamp and MTs onto Cryo-EM grids. Line 806-811: The method described is unclear. Does it mean that the 9 μM tubulin was diluted in equal volume to 4 μM N-clamp, resulting in a mixture containing 4.5 μM tubulin heterodimer and 2 μM N-clamp; to which an additional 4 μl of 4 μM N-clamp was added? This should probably be re-written to assist the reader.

We apologize and have rephrased the methods section for cryo-EM grid preparation of the wild-type N-clamp. We have furthermore included a section for cryo-EM grid preparation of the GST N-clamp, which followed the same principle:

“Cryo-EM sample preparation

MTs pre-incubated with *D. melanogaster* augmin N-clamp

After AEC, purified wild-type *D. melanogaster* augmin N-clamp was desalted in BRB80 using a HiTrap 5ml column followed by concentration using Amicon, 30 kDa MWCO (Merck) to a final concentration of 4 μM . Taxol-stabilized MTs were prepared as described above and diluted to a concentration of 1 mg/ml in BRB80 buffer at RT. Before plunge freezing taxol-stabilized MTs were mixed in a 1:1 ratio with the desalted *D. melanogaster* augmin N-clamp (4 μM concentration). Holey carbon grids (Cu R2/1, 200 mesh; Quantifoil) were glow-discharged twice for 1 min using an easiGlow plasma cleaner (PELCO). 3 μl of the incubation mixture was applied on the carbon side of the EM grids mounted into a Vitrobot Mark IV (Thermo Fisher Scientific) operated at 30°C and 100% humidity. After 30 sec of incubation, 4 μl of purified *D. melanogaster* augmin N-clamp sample (4 μM concentration) was added to maximize binding. After a waiting time of 1 min and 40 s, the grids were blotted for 1 s from both sides and plunge-frozen in liquid ethane. The grids were screened at a Glacios TEM (Thermo

Fisher Scientific) to assess their quality and representative images were collected at different magnifications.

MTs decorated with *D. melanogaster* augmin N-clamp on grid

Purified *D. melanogaster* augmin N-clamp and MTs were prepared as described above (pre-incubation). Holey carbon grids (Cu R2/1, 200 mesh; Quantifoil) were glow-discharged twice for 1 min using an easiGlow plasma cleaner (PELCO). 3 μ l of MT solution (1 mg/ml) was applied on the carbon side of the EM grids and MTs were allowed to adsorb for 1 min at RT and ambient atmosphere before a short manual blotting step (Whatman filter paper #1). Subsequently, the grids were incubated with 3 μ l of purified *D. melanogaster* augmin N-clamp sample (4 μ M concentration), waiting 2 min and manually removing excess buffer (Whatman filter paper #1) between each application. Afterwards, the grids were mounted into a Vitrobot Mark IV (Thermo Fisher Scientific) operated at 30°C and 100% humidity. Finally, 4 μ l of purified *D. melanogaster* augmin N-clamp sample (4 μ M concentration) was added to maximize binding. After a waiting time of 1 min and 40 s, the grids were blotted for 1 s from both sides and plunge-frozen in liquid ethane. Grids were stored in liquid nitrogen until further use.

MTs decorated with *D. melanogaster* augmin GST-N-clamp on grid

MTs were prepared as described above and diluted to a final concentration of (0.5 mg/ml). Purified *D. melanogaster* augmin GST-N-clamp was desalted and concentrated as described above to a final concentration of 4 μ M. Holey carbon grids (Cu R2/1, 200 mesh; Quantifoil) were glow-discharged twice for 1 min using an easiGlow plasma cleaner (PELCO). 3 μ l of MT sample (0.5 mg/ml) sample was applied on the carbon side of the EM grids and MTs were allowed to adsorb for 1 min at RT and ambient atmosphere before a short manual blotting step (Whatman filter paper #1). Subsequently, the grids were incubated four times with 3 μ l of purified *D. melanogaster* augmin GST-N-clamp sample (4 μ M concentration), waiting 30 sec and manually removing excess buffer (Whatman filter paper #1) between each application. Afterwards, the grids were mounted into a Vitrobot Mark IV (Thermo Fisher Scientific) operated at 30°C and 100% humidity. Finally, 4 μ l of purified *D. melanogaster* augmin GST-N-clamp sample (4 μ M concentration) was added and after 30 sec the grids were blotted for 1 s from both sides and plunge-frozen in liquid ethane. Grids were stored in liquid nitrogen until further use.”

Additional minor comments:

Line 218 – remove the word “essential”. The authors write that the CH6 domain is “a conserved and essential MT-binding element”. Yet, it is clear from the data (Figure 3c) – and is stated in the text - that, in the absence of CH6 in humans, the N-clamp still binds MTs; just a lower fraction.

We have adapted the text as suggested.

Supplementary Figure 9 – in (d), Dtg2 should be Dgt2

We have corrected the labeling.

Reviewer #2 (Remarks to the Author):

This manuscript focuses on binding mechanism of the microtubule-interacting region of the augmin complex known as the N-clamp and associated implications for microtubule branching. Using structural prediction and co-sedimentation assays, this work identifies the minimal microtubule-binding region of augmin across four species to be the HAUS6 CH domain, identifying more or less contribution to binding of accessory regions, in particular the disordered HAUS8 N-terminus. The *Drosophila* N-clamp was further investigated with mutational and structural analysis, as it lacks the HAUS7 CH and has a particularly short HAUS8 N-terminus, such that HAUS6 CH is the key mediator of microtubule interaction. AlphaFold predictions combined with cryo-EM and molecular dynamics analysis suggest a model for HAUS6 CH interaction with the interprotofilament groove between beta-tubulins. Finally, the HAUS6 CH was shown to be of likely importance in the binding of human augmin to microtubules, based on mutagenesis studies on mitotic spindle architecture in cells.

A key finding of this work is the importance of the HAUS6 CH domain in augmin microtubule interaction across species and the variable contribution of the HAUS8 N-terminus, which was previously thought to be more central to the interaction (although it is required in humans).

#1 While the non-structural parts of the study are sound, my main issue with this work is that the cryo-EM analysis is not of sufficient quality to unambiguously define the binding mode of the Dm HAUS6 CH. Without sufficient improvement in the quality of cryo-EM density for the HAUS6 CH, the study is too reliant on AlphaFold predictions and not suitable for publication.

To address this comment, we performed extensive additional cryo-EM experiments (see below for details) that ultimately improved resolution of our cryo-EM reconstruction to secondary structure level for the CH6 domain and allowed unambiguous rigid body docking of the AF2 prediction into our improved density.

First, we tried to improve the quality of our cryo-EM reconstruction by doubling the size of our cryo-EM dataset (from ~9k to 18k micrographs). Additionally, using our extended cryo-EM dataset, we closely followed the image processing approaches outlined in the studies that Reviewer #2 provided for reference (for example: Cook et al., 2020, or Atherton et al., bioRxiv 2024 (<https://doi.org/10.1101/2024.11.11.622991><<https://doi.org/10.1101/2024.11.11.622991>>)). This included density subtraction, extended local refinements and recentering of cryo-EM particles in the asymmetric unit. However, none of these alternative image processing strategies led to a substantial improvement of the quality for the decoration density. Of note, the CH6 of the N-clamp construct is substantially smaller and has a much more limited binding interface towards MTs, as compared to the MT-binding proteins studied in the literature provided by Reviewer #2 (Kinesins). This suggested that either the small size and limited binding interface of the N-clamp, or the comparatively low apparent MT binding affinity of the N-clamp (~ 0.4 μ M) was limiting the attainable resolution.

To address the problem of limited affinity while preserving the native N-clamp-MT interaction, we generated a dimerizing GST-tagged *D. melanogaster* N-clamp version

(GST-N-clamp). Complex formation and GST-induced dimerization were confirmed by mass photometry and tubulin co-sedimentation assays showed that GST-mediated dimerization decreased the apparent K_d approximately 10-fold to ~40 nM. We then repeated cryo-EM experiments with the GST-N-clamp analogously to wild-type N-clamp. Following a comparable image processing approach, we could obtain a cryo-EM reconstruction of the MT-bound N-clamp in which secondary structure elements of the CH6 domain could be unambiguously assigned and density for more distal segments of the N-clamp were visible at lower threshold levels (Supplementary Fig. 10). In addition, while unsuccessful for the wild-type N-clamp, the image processing approaches outlined in literature suggested by Reviewer #2 further improved particle sorting and the quality of the final cryo-EM reconstruction to secondary structure level. This enabled highly-confident docking of the N-clamp AF2 model and at the same time confirmed that AF2 correctly predicted the fold of the CH6 domain. Importantly, the positioning of the CH6 domain in our cryo-EM reconstructions of wild-type and GST-N-clamp was highly comparable, in particular with respect to the binding interface between CH6 and the MT lattice.

Collectively, using a GST N-clamp variant with 10-fold higher MT-binding affinity allowed us to substantially improve our cryo-EM reconstruction, which led to a major advance in the interpretability of our cryo-EM data.

#2 Obviously, the CH density being at near-atomic resolution would be the ideal situation, but this is not always possible. However, given the dataset size and the lack of flexibility in the CH indicated by AlphaFold and molecular dynamics, I don't see why this density should not be at a higher resolution.

As outlined in our response to main comment #1 above, we were able to substantially improve our cryo-EM reconstruction by generating a dimerizing N-clamp variant with 10-fold improved MT binding affinity. This enabled us to resolve the individual alpha-helices of the CH6 domain and allowed for highly confident rigid body docking of the CH6 atomic model predicted by AF2. Sample-inherent challenges, such as the small size, the limited MT binding interface and remaining conformational flexibility may limit the attainable resolution to secondary structure level even for our improved GST N-clamp variant.

#3 With a good sample (including good decorating protein occupancy) this microtubule processing pipeline should be producing higher resolution details at least with the version operating in Relion v3.0 (for example see Cook et al., 2020, or Atherton et al., bioRxiv 2024

(<https://doi.org/10.1101/2024.11.11.622991> <<https://doi.org/10.1101/2024.11.11.622991>>).

We thank the reviewer for the suggestion. We closely followed the processing approaches outlined in these references. While we could achieve no improvement for the decorating density in case of wild-type N-clamp, local resolution and density quality was substantially improved for the GST-N-clamp and the respective steps were therefore included in the final image processing scheme (Supplementary Fig. 10).

#4 The CH required low-pass filtering, which makes unambiguous fitting (even rigid-body) challenging for a small globular domain. Indeed, looking at the provided reconstructions, the alpha-beta tubulin register is not particularly well resolved and the density for the CH domain only appears at low threshold. The fit is not great even at this resolution, particularly for the two longest helices in the CH domain. This raises the possibility of some inaccuracy in the alphafold predictions, or a conformational change upon microtubule association that has not been modelled.

Using a GST N-clamp variant with 10-fold higher MT binding affinity and an extended image processing scheme suggested by Reviewer #2, we could substantially improve our cryo-EM reconstruction to resolve the CH6 domain to secondary structure level. This enabled unambiguous rigid body docking of the AF2 model for the N-clamp and confirmed that AF2 was capable of correctly predicting the fold of the CH6 domain.

#5 My feeling from the data provided and considering the concentrations of N-clamp used is that sub-stoichiometric occupancy of the N-clamp on the microtubule may be an issue in this dataset and/or suboptimal image processing.

As described in more detail in our response to main comment #1 above, using a GST N-clamp variant with 10-fold higher MT binding affinity enabled us to substantially improve our cryo-EM reconstruction, suggesting that indeed sub-stoichiometric occupancy was a major challenge when using the wild-type N-clamp for cryo-EM sample preparation.

#6 If the authors can improve the cryo-EM reconstructions/model in a revised submission, it would be suitable for publication.

As requested by Reviewer #2, we have substantially improved our cryo-EM reconstruction in the revised manuscript by extensive additional cryo-EM experiments. Our improved cryo-EM reconstruction now enables unambiguous rigid body docking of the AF2 model for the N-clamp and confirms that AF2 was capable of correctly predicting the fold of the CH6 domain.

Other specific comments;

1) Sup Fig 3 c is a little unconvincing. A higher mag view of the sample would be more useful.

We added a higher magnification view of the micrograph.

2) Line 145 should also refer to Figs showing co-sedimentation raw data examples.

We have added the reference to the raw data.

3) Line 227 is worded in such a way it seems there may have been some augmin-mediated nucleation of MTs from pre-existing taxol-stabilised MTs. This is not the case and this should be made more clear.

We have adapted the text.

4) Line 228- could the authors comment on whether the bundling represents any feature of biological interest?

We thank the reviewer pointing to this aspect. As we don't have direct evidence to judge if it represents a biological feature or an artifact of microtubule decorating proteins, we included this aspect in the discussion section:

“Notably, we observed pronounced MT bundling activity of our N-clamp construct *in vitro* (Supplementary Fig. 7), consistent with previous reports for HAUS8 (formerly HICE1) ³⁶. Although MT bundle formation has been reported *in vivo* in the context of kinetochore fibers and dense spindle arrays ¹³, whether augmin directly contributes to this organization through bundling activity is unknown. In fact, MT bundling is a common feature of multivalent MT-binding proteins *in vitro* ^{64,65} and may arise from high local protein concentrations or other non-physiological experimental conditions. Addressing whether the MT bundling effect of the augmin N-clamp reflects a specific biological function will require further investigations. “

5) Line 237. EM analysis. The local resolution shown indicates that the resolution of the N-clamp is ~ 6-10Å. Why then is a 15Å low pass filter applied?

When filtered to the estimated local resolution, the decorating density was still partially fragmented and visualization was challenging. Using our GST-N-clamp with vastly improved MT affinity, we now obtained a reconstruction in which none of the problems persist. For details, please see our response to major comment #1 above.

6) Figures 4b/c should be shown with taxol model for full assessment of resolving tubulin register.

We have updated the figure panels as suggested.

7) Line 253. In Fig sup 7e we can see some of the rest of the N-clamp in the 2D classes. This makes it strange that we can't see much in 3D. Why might this be? Does it appear at different thresholds or with different filtering?

In our original cryo-EM reconstruction of wild-type N-clamp, density for more distal sections was visible only at very low-threshold levels. In the cryo-EM reconstruction of our improved GST-N-clamp variant, these more distal sections appear already at higher threshold levels and are better defined as in the reconstruction of wild-type N-clamp, as shown in *Supplementary Fig. 10c* of the revised manuscript. Still, the more distal segments of the N-clamp are not equally well represented as CH6 in the cryo-

EM reconstruction, suggesting some remaining conformational flexibility, consistent with our MD simulations.

8) Line 303. I'd dispute the high similarity- there is quite a bit of the model outside the density!

While the overall shape of the decorating density in the original cryo-EM reconstruction was generally compatible with the CH6 domain structure, we agree that it did not optimally cover some individual α -helices in the AF2 model of the CH6 domain. Importantly, our new cryo-EM reconstruction obtained using GST-N-clamp resolves the CH6 domain at secondary structure level and matches the AF2 model very well, enabling highly confident rigid body docking. For details, please refer to major comment #1 above.

9) Cryo-EM methods section- please indicate final concentrations of N-clamp used in sample prep more clearly.

We have adapted the methods section.

10) What was the reference used during cryo-EM microtubule filament refinements? This is very important for understanding what has happened during processing.

As reference for the initial particle refinements, we used the final 3D class of the MiRP pipeline before seam correction. For subsequent steps, densities of preceding 3D refinement runs were used as references. We now state this more clearly in our cryo-EM method section.

Reviewer #3 (Remarks to the Author):

I am a specialist in atomistic molecular dynamics simulations, and I have also used these in conjunction with cryo-EM data. While I do have an interest in molecular motors and microtubules, this is not my main area of expertise. I am not an expert in the experimental protocols used in the paper so I will not comment on their validity. I will also leave it to the experts on the cytoskeleton to comment on whether this publication introduces sufficiently new information compared to Zupa et al Nat Comms 2022 to be novel enough to merit another Nat Comms paper, however, from my perspective the additional atomistic detail about the binding interface does merit publication in a high impact journal.

We thank the reviewer for the positive assessment and agree that dissecting the exact mechanism of how augmin associates with MTs represents an important advance in our mechanistic understanding of several central cellular processes.

#1 The use of alpha-fold models to build the initial structures for fitting into the cryo-EM density is very sensible, and it is interesting to compare across different species.

I am presuming the authors chose the *Drosophila* structure because it does not contain as many disordered regions in the alpha-fold model as the other species they modelled. Please can the authors clarify this in the paper.

Yes, we chose *D. melanogaster* because of the comparably simple architecture of its MT binding interface, lacking the CH7 domain and the extensively long flexible N-terminus of HAUS8. We clarified this in the manuscript:

“To test the function of the conserved CH6 residues in a low-complexity system without interference from other MT-binding N-clamp elements, we analyzed *D. melanogaster* augmin N-clamp, where the CH7 domain is absent and the Dgt4_{HAUS8}-NT deletion had little impact on MT binding (Fig. 2d).”

“Next, we aimed at elucidating the structural basis for how CH6 anchors and orients augmin on MTs using cryo-EM. To this end, we used the *D. melanogaster* augmin N-clamp as a low-complexity model system, in which CH6 plays a dominant role in the augmin-MT interaction, while the CH7 domain is absent and the Dgt4_{HAUS8}-NT deletion had little impact on MT binding.”

#2 I also became slightly confused about the E-hook and the missing loop region in *Drosophila* relative to the other species and whether it is present or absent in the structure/sequence. I read the text as meaning it is absent from both structure and sequence in the fly protein, but in Fig 1b it does look like there could be missing residues in the circled area (but clearly it is not possible for the reader to know this for certain without checking the sequence).

In the revised version of the manuscript, we clarified this aspect. We included additional experiments (see *Reviewer 1, comment #4*) and a more extended discussion section addressing the interaction between the *Msd5*_{HAUS7} loop and the tubulin E-hooks, including graphical representations of the loop (Supplementary Fig. 14). We also modified the legend for Fig. 1b to point out more clearly that the circled region refers to the CH domain of *Msd5*_{HAUS7} and not the loop.

#3 In their MD methods, please can the authors state whether the termini of protein chains are charged or neutralised with capping groups in their simulations, and why explain why this decision was made.

We chose to model the N- and C-termini as charged because they are located in flexible loops exposed to solvent. Additionally, AlphaFold prediction showed that the N-terminal groups of both the N-clamp and tubulin dimers are in close proximity. Given their charged nature, we hypothesized that they could contribute to stabilizing or destabilizing the interaction interface. We have added a clarification to the Methods section and thank the reviewer for bringing it to our attention.

#4 Could they also make it clearer whether the production run MD is unrestrained, or whether they have kept restraints on the backbone and/or sidechains.

The production run MD simulations include positional restraints, but only on the two lateral rows of tubulin dimers. These restraints are applied to the heavy atoms to preserve the overall shape of the microtubule without requiring simulation of the entire tube. The system consists of four rows of tubulin dimers: the two lateral rows are restrained, while the two central rows, those interacting with the N-clamp, are left unrestrained. This setup preserves realistic dynamics and allows for an unbiased interaction between tubulin dimers and the N-clamp. We have now updated the Methods section to clarify this point.

#5 Could they also please explain why only the final 200ns was used in the analysis and why this segment was chosen - for example did they see significant structural relaxation in the first 800ns of the trajectory?

The three simulation repeats did not show significant differences in the overall interaction interface. The interface appeared very stable across all three trajectories. Therefore, we treated the first 800 ns as equilibration time, allowing the system to relax, and focused our analysis on the final 200 ns, during which the system remained largely stable, as confirmed by the RMSD we have now added as new Supplementary Figure 11a. We have now added the contact map, interaction energies, RMSF, and a visual representation of the representative structure of the MT-bound augmin N-clamp, recalculated over the last 500 ns, as new Supplementary Figures 11, 13, 15. The key interacting residues remain present, although some contacts disappear from the map because their frequencies dropped below our 0.5 threshold.

#6 Please could they also explain what is measured by the function `gmx_energy` and justify the use of this measurement in their analysis. Energy calculations that involve electrostatics are subtle in MD simulations, because the presence of the solvent screens the charges, so often implicit Poisson-Boltzmann models (or approximations thereof) are used.

The use of `gmx energy` in GROMACS allows us to extract non-bonded interaction energies, specifically Coulomb (electrostatic) and Lennard-Jones (van der Waals) contributions, between defined groups of atoms, based on atomic positions and force field parameters at each simulation time step.

While we recognize that electrostatic interactions in solvated systems are influenced by dielectric screening, and that continuum electrostatics models (e.g., Poisson-Boltzmann) can offer more quantitative insight, our goal here is qualitative. Specifically, we use interaction energy calculations to refine our interpretation of the contact map. We wanted to distinguish residues that are truly involved in energetically significant interactions from those that are merely in spatial proximity to interacting residues. This helps avoid overinterpretation of purely distance-based contacts and allows a refined characterization of the interaction interface.

This qualitative approach is consistent with our previous work (Steringer et al., eLife, 2017; Lagrande et al., Communications Biology, 2020), where classical force field-derived energies were used to support contact-based structural analyses.

We have now added a paragraph in the Methods section to address this point.

#7 Have the authors considered sharing a pdb file containing the structures used to run their simulations, as well as the truncated structure (model.cif) in their zip file containing the cryo-EM maps and models?

Yes, we have provided the Zenodo link in the Data Availability statement, where we included all the files necessary to run the simulations, as well as the trajectories. We have just realized that the original upload only contained GROMACS-formatted files. We have now updated the link to also include the corresponding PDB file:

<https://doi.org/10.5281/zenodo.15492509>

#8 What differences (if any) did the authors notice between their independent repeat simulations?

As shown by the RMSD, the docked structure, generated by AlphaFold and fitted into the cryo-EM density, was surprisingly stable from the beginning, without requiring major rearrangements. This stability is further supported by our interface analysis, which reveals a robust network of ion-pair interactions involving the N-clamp. Importantly, these interactions were consistent across simulations and not affected by the randomized initial conformations. Only in one simulation did we observe a minor loop rearrangement, which did not significantly contribute to the overall interaction network.

#9 Overall I would assess the simulations that have been run as high quality, given that there are 3 independent repeats, and that the simulation timescales correspond to current state of the art (although the 200ns segment used to perform the analysis is somewhat shorter than usual).

We would like to thank the reviewer for their supportive comments and constructive feedback. As requested, we have now extended the simulation time used for the analysis and report as new Supplementary Figures 11, 13, 15.

#10 I would have liked to be able to look at the structures sampled during the MD, however the movie provided as supplementary information was useful.

In the Data Availability statement, we had included a Zenodo link, which provides access to the initial configuration files for the Molecular Dynamics simulations associated with this study, as well as the corresponding trajectories. We have now updated the repository to also include the structure file in PDB format: <https://doi.org/10.5281/zenodo.15492509>

We thank the Reviewers for their thoughtful and supportive comments. Below we address the remaining points and elaborate on the corresponding changes in the manuscript.

REVIEWER COMMENTS

Reviewer #1 (Remarks to the Author):

I am pleased that the authors have comprehensively addressed all my comments, and have undertaken additional CryoEM studies that enhance their conclusions. This further strengthens the manuscript and I'm happy to recommend it for publication.

We thank Reviewer #1 for the positive assessment of the revised manuscript.

Reviewer #2 (Remarks to the Author):

Lots of hard work has been done to improve the reconstruction via a combination of engineering a dimer for better affinity/occupancy and improvements in the image processing methods. The rigid fitting now looks way more convincing for the N-clamp, particularly the CH domain Haus6 region.

We thank Reviewer #2 for the positive evaluation of the improved cryo-EM reconstruction.

The authors have addressed my main concern here, but I would like to make two suggestions- 1) is to show the same model in the Figures as has been provided. For example, Haus8's fitted helix is longer in figure 4 than it is in the deposited model (see attached images of figure 4 vs a chimera session with the model/map). The angle of the image suggests a good fit, but I expect this is an exaggeration.

We thank Reviewer #2 for this suggestion. We have updated Figure 4 accordingly.

2) I would say the helices in the non-CH domain region of the N-clamp (Haus 8/7 and one from HAUS6) could arguably be extended and flexibly fitted into the map- it seems like their orientation may be different to the alphafold model and this could be interesting for functional interpretation. It could also just be that the map is poorer/lower resolution in this region because of flexibility- which fits with the MD simulations- hindering a good fit. I do understand the hesitation to flexibly fit even at this resolution- so I would suggest that the authors just mention/discuss the not-so-good fit of these regions relative to the core of the CH domain.

We fully agree with Reviewer #2. The more distal regions of the augmin N-clamp were not sufficiently resolved to warrant flexible fitting of the atomic model predicted by AlphaFold2. As suggested, we have added the following sentence to the result section:

“Notably, MT-proximal α -helices of Msd5_{HAUS7} and Dgt4_{HAUS8} were slightly differently oriented between the cryo-EM reconstruction and the AF2 prediction, which may

reflect local inaccuracies of the model predicted by AF2 or suggest small conformational rearrangements of the augmin N-clamp upon MT binding.”

Reviewer #3 (Remarks to the Author):

The authors have carefully responded to their comments from the referees, and I am satisfied that the simulations described in the manuscript are of high quality. I would suggest trying more sophisticated energy calculations (e.g. MMPBSA) when assessing energy decompositions in the future, to see if that provides any additional insight.

We thank Reviewer #3 for this valuable suggestion for follow-up experiments.

I presume that the simulations are the same as previously and have not been rerun following the improved structure determination - please can the authors justify this decision in their subsequent response to reassure the editor that this is appropriate.

Yes, the MD simulations have not been rerun based on the improved cryo-EM reconstruction. When comparing the position and conformation of the CH6 domain in the improved cryo-EM reconstruction and the final frame of our MD simulation, we observed excellent agreement between the two models. This confirmed that the slight reorientation of the CH6 domain in the improved cryo-EM reconstruction was in a range that was easily captured during the course of the simulation, clearly indicating that a repetition of the MD simulations with new starting coordinates was not necessary.